# MULTI-SESSION, MULTI-TASK NEURAL DECODING FROM DISTINCT CELL-TYPES AND BRAIN REGIONS

**Mehdi Azabou**[*]
Georgia Tech

**Krystal Xuejing Pan**[*]
Mila

**Vinam Arora**
Georgia Tech

**Ian Knight**
Georgia Tech

**Eva L. Dyer**[†]
Georgia Tech

**Blake Richards**[†]
Mila, McGill University

## ABSTRACT

Recent work has shown that scale is important for improved brain decoding, with more data leading to greater decoding accuracy. However, large-scale decoding across many different datasets is challenging because neural circuits are heterogeneous—each brain region contains a unique mix of cellular sub-types, and the responses to different stimuli are diverse across regions and sub-types. It is unknown whether it is possible to pre-train and transfer brain decoding models between distinct tasks, cellular sub-types, and brain regions. To address these questions, we developed a multi-task transformer architecture and trained it on the entirety of the Allen Institute's Brain Observatory dataset. This dataset contains responses from over 100,000 neurons in 6 areas of the brains of mice, observed with two-photon calcium imaging, recorded while the mice observed different types of visual stimuli. Our results demonstrate that transfer is indeed possible—combining data from different sources is beneficial for a number of downstream decoding tasks. As well, we can transfer the model between regions and sub-types, demonstrating that there is in fact common information in diverse circuits that can be extracted by an appropriately designed model. Interestingly, we found that the model's latent representations showed clear distinctions between different brain regions and cellular sub-types, even though it was never given any information about these distinctions. Altogether, our work demonstrates that training a large-scale neural decoding model on diverse data is possible, and this provides a means of studying the differences and similarities between heterogeneous neural circuits.

## 1 INTRODUCTION

All behaviors, perceptions, and thoughts arise from the activity of neural circuits in the brain. In principle, decoding this neural activity could allow us to infer what someone is doing, perceiving, or thinking. Effective decoding of neural activity holds the potential to unlock transformative applications, such as brain-computer interfaces for thought-controlled systems and advanced clinical interventions like targeted deep brain stimulation (Saha et al., 2021; Merk et al., 2022).

Recent advancements in transformer architectures (Radford et al., 2018) have introduced a new paradigm in brain decoding (Azabou et al., 2024; Ye et al., 2024; Zhang et al., 2024). These models demonstrate that increasing the scale of neural data, by training on many recording sessions and across multiple animals, leads to significant improvements in decoding performance. This trend mirrors the success of large-language models, where pretraining on large corpora has resulted in models capable of generalizing across a wide range of tasks. In the case of brain decoding, larger datasets allow models to capture more nuanced patterns of neural activity, enabling better generalization to new behaviors (Azabou et al., 2024; Jiang et al., 2024). This scaling effect is especially evident when small, previously uninformative datasets are paired with pretrained models, unlocking new decoding abilities and overcoming limitations imposed by insufficient data (Azabou et al., 2024).

---

[*]These authors contributed equally to this work [†] These senior authors contributed equally to this work. Contact: mazabou@gatech.edu. Project page and code: `https://poyo-plus.github.io`

However, large-scale training for brain decoding is challenging due to the diversity of neural recordings. Though there is a lot of neural data available in aggregate, it is mostly small, highly heterogeneous individual datasets (Ferguson et al., 2014). Each individual dataset can contain recordings that involve distinct cellular sub-types and brain regions - as well as different stimuli or behaviors (Esfahany et al., 2018; Dipoppa et al., 2018; Scala et al., 2021). This diversity in data complicates efforts to train large-scale models on large corpora. Prior work has demonstrated that scaling data within neural populations that are relatively similar across sessions can improve decoding accuracy (Azabou et al., 2024; Ye et al., 2024; Zhang et al., 2024), but it is unknown whether similar improvements can be achieved when training models on *heterogeneous* datasets involving different cellular sub-types, brain regions, and tasks. Can neural decoding models be trained on multiple decoding tasks at once? Can neural decoding models trained on one cellular sub-type or brain region transfer to another? Can pretraining on diverse data yield models that generalize across distinct neural circuits, or does this diversity introduce too much noise for effective learning?

To address these open questions, we developed a multi-task, multi-modal transformer architecture and trained it on the Allen Institute for Brain Science Observatory dataset. This dataset includes recordings from over 110,000 units recorded from mice using two-photon calcium imaging while they observed various types of visual stimuli (de Vries et al., 2020). These recordings span 256 animals, 6 brain regions, and 13 genetically defined cellular sub-types, making it one of the most diverse neural data archives available. By leveraging this large and heterogeneous corpus of data, we aimed to test whether transfer is possible across distinct brain regions and cellular sub-types, and whether pretraining on diverse data can improve decoding performance for a variety of tasks.

Our results demonstrate that training across brain regions and cellular sub-types—-despite their functional differences—-is not only feasible but also beneficial. We show that combining data from heterogeneous recordings leads to improved performance on downstream decoding tasks, indicating that there is shared information across diverse neural circuits that can be captured by an appropriately designed model. Notably, even in cases where single-session baselines fail to decode stimulus information, pretraining on diverse datasets significantly enhances decoding capabilities. This finding suggests that neural circuits, despite their heterogeneity, share common computational principles that can be harnessed by a large-scale model for decoding.

We then examined our model's latent embeddings and discovered that they revealed meaningful structure. Even though the model was not trained to differentiate brain regions or cellular sub-types (and received no information about this), its latent space naturally organized neurons based on their anatomical and physiological similarities. When we clustered the latent representations, we observed that cellular sub-types and brain regions with more similar anatomy and physiology were more closely aligned in the latent space. This emergent organization provides new insights into the relationships between different neural circuits and suggests that large-scale models can capture higher-order patterns of neural activity that reflect the underlying functional architecture of the brain.

In summary, the main contributions of this work include:

- **Multi-Task and Multi-Modal Decoder for Flexible Querying:** We introduce a novel multi-task decoder that can query the learned latent space to engage in a wide range of decoding tasks with high fidelity. This decoder is capable of handling both continuous and discrete decoding tasks, demonstrating versatility across different neural recordings and experimental setups.

- **Cross-Region and Cross-Cell-Type Transfer:** Our model shows successful transfer learning across brain regions and genetically distinct cellular sub-types. This cross-area and cell line transfer demonstrates that the model can generalize representations beyond specific regions and sub-types, capturing shared computational principles across diverse circuits.

- **Across-Region Scaling:** We provide evidence that increasing the diversity of brain regions in the training data leads to improved decoding performance. As more regions are included in training, the model's ability to decode and generalize across previously unseen regions significantly improves.

- **Meaningful Latent Representations:** We show that the model's latent representations of the data carry meaningful information that reflect the anatomy and physiology of different regions and sub-types. This illustrates that it is possible for large-scale decoding models to identify biologically meaningful distinctions from neural activity alone.

Figure 1: **Our approach to diverse multi-region, multi-task neural decoding.** Our approach leverages a task embedding to query the neural data latents to decode different contextual or behavioral information from the neural activity.

## 2 METHODS

In this section, we describe POYO+ , our multi-task, multi-modal model for neural decoding (illustrated in Figure 1). The main innovation of our approach is the development of a multi-task decoder that allows for flexible training with different stimuli and behavioral measures as decoded outputs, across varying contexts, all through a set of learned task embeddings.

### 2.1 TOKENIZATION

The calcium recordings consist of multiple time-series, one for each neuron that was identified in the calcium imaging session. The values in the time-series correspond to the change in fluorescence over a baseline value (i.e. $\Delta F/F$), which reflects the in action potential firing of a neuron. Let $\mathbf{f}_{ij}$, denote the scalar value of the original fluorescence signal, $\Delta F/F$, at time-step $i$ for neuron $j$.

To tokenize the activity of a population of $P$ neurons, we adapt the POYO tokenizer (Azabou et al., 2024) to regular timeseries data by creating a token for each timestep and for each neuron, i.e. over $T$ time-steps we create a sequence of $P \times T$ tokens $(\mathbf{x}_{ij}, t_{ij})$, where $i \in [1, \ldots, T], j \in [1, \ldots, P]$. These tokens carry information about the neuron's activity, which in turn can be used to learn about it other properties—e.g., cell-type, cortical layer, and the brain-region to which it belongs. Specifically, each token, is a tuple with two components: $(\mathbf{x}_{ij}, t_{ij})$, where $\mathbf{x}_{ij} = [\mathbf{u}_j, \mathbf{W}_f \mathbf{f}_{ij}]$ is an embedding consisting of information about the unit and signal amplitude determined by learnable parameters $\mathbf{W}_f \in \mathbb{R}^N$. $\mathbf{u}_j$ is a $N$-dimensional, neuron-level learned, embedding that is constant across time-steps. Each neuron receives its own learned embedding, which is accessed via a look-up table. Importantly, $\mathbf{u}_j$ does not contain any information about the brain region the recording was done in, nor the cellular sub-type of the neuron, prior to learning. $t_{ij}$ is a $2N$-dimensional sinusoidal time-embedding, which encodes the timing of the token relative to the start of the context window. In this way, the activity of any population of neurons can be represented flexibly through a single sequence of tokens, which is passed as input to the transformer model.

### 2.2 POYO ENCODER

To build rich representations of neural activity, we use the POYO encoder (Azabou et al., 2024) which employs a cross-attention layer to compress neuron-level spike tokens into a lower-dimensional latent space. In our case, we will learn a set of $M \times N$ latent token tuples $((\mathbf{z}_{mn}^0, \tau_{mn}),$ $m \in [1, \ldots, M], n \in [1, \ldots, N])$ that will be used to generate queries for the neural data. Here, each $\mathbf{z}_{mn}^0$ is a learned latent vector, and $\tau_{m,n}$ is sinusoidal rotary embedding. The index $m$ is used to indicate different latent tokens, and the index $n$ is used to indicate the different time-steps across the context window, which are evenly spaced. The cross-attention operation is given by:

$$\mathbf{z}_{mn}^1 = \mathbf{z}_{mn}^0 + \sum_{i=1}^{T} \sum_{j=1}^{P} \text{softmax}\big( \left(\mathbf{R}(\tau_{mn})\mathbf{q}_{mn}\right)^T \left(\mathbf{R}(t_{ij})\mathbf{k}_{ij}\right) \big) \mathbf{v}_{ij}, \tag{1}$$

where the values $\mathbf{v}_{ij} = W_V^0 \mathbf{x}_{ij}$ and keys $\mathbf{k}_{ij} = W_K^0 \mathbf{x}_{ij}$ are computed from the neural activity tokens, and the queries $\mathbf{q}_{mn} = W_Q^0 \mathbf{z}_{mn}^0$ are computed from the learned latent tokens, and $\mathbf{R}(t)$ is a predefined rotation matrix whose values are determined by the time-step given (see Appendix C.1). Rotating keys and queries (Su et al., 2021) allows us to incorporate information about the relative

timing between two tokens into the attention mechanism. As mentioned above, our latent tokens are positioned uniformly across the context window to provide "virtual timesteps" for each query.

After cross-attention, we then apply $L$ rounds of self-attention to compute a sequence of final latents $\mathbf{z}_{mn}^L$. Throughout, we use rotary time encoding with standard transformer blocks, pre-normalization layers and feed-forward nets.

## 2.3 MULTI-TASK, MULTI-MODAL DECODER

Neural datasets often have multiple sources of behavioral information that are simultaneously acquired along with brain activity (i.e., pupil diameter, running speed) while different stimuli are presented to the animal. This diversity in potential decoding targets must be harnessed as we scale and build models that can generalize to diverse datasets.

To accommodate this type of diversity, we developed a model that can be trained jointly on different measures of behavior as well as information about the stimuli presented or context of the recording (Figure 1). In order to have a generalist model that can work with these different types of labels, we built a flexible decoder that uses a shared cross-attention layer to query from the latent space using different task-specific tokens. This allows us to train on heterogeneous neural datasets and decode any number of different variables without assuming the same behaviors or stimuli are present in the recordings. Indeed, we can combine both dense, continuous regression decoding tasks (like pupil diameter prediction or running speed prediction), sequence-level classification tasks (like drifting gratings orientation classification or static gratings spatial frequency classification), and segmentation tasks (like movie frame prediction or locally sparse noise pattern prediction).

Our multi-modal decoder uses a shared cross-attention layer to query the latent space for labels relevant to the different downstream tasks. An output query token for decoding task $k$, in recording session $\ell$, at time-step $t_i$, is a tuple $(\mathbf{o}_{\ell k}, t_i)$. The component $\mathbf{o}_{\ell k} = \phi_k + \mathbf{s}_\ell$ is defined by two quantities: (i) **Task embedding:** $\phi_k \in \mathbb{R}^D$ is a learned embedding corresponding to the decoding task of interest (e.g. running speed regression or drifting grating orientation classification, etc.). As with neurons, each task receives its own learned embedding that is accessed via a look-up table; (ii) **Session embedding:** $\mathbf{s}_\ell \in \mathbb{R}^D$ is a learned embedding corresponding to the recording session. Again, each recording session receives its own learned embedding accessed via a look-up table. Session embeddings are used to capture variability across sessions in the datasets.

We calculate the final output tokens for the ith time-step, $\mathbf{y}_{\ell k i}$, using a cross-attention mechanism:

$$\mathbf{y}_{\ell k i} = \mathbf{o}_{\ell k} + \sum_{n=1}^{N} \sum_{m=1}^{M} \mathrm{softmax}\big( \left(\mathbf{R}(t_i)\mathbf{q}_{\ell k}\right)^T \left(\mathbf{R}(\tau_{mn})\mathbf{k}_{mn}\right) \big) \mathbf{v}_{mn}, \tag{2}$$

where we have queries $\mathbf{q}_{\ell k} = W_o \mathbf{o}_{\ell k}$, values $\mathbf{v}_{mn} = W_V^L \mathbf{z}_{mn}^L$, and keys $\mathbf{k}_{ij} = W_K^L \mathbf{z}_{mn}^L$, calculated from our final set of tokens produced by self attention. As before, $\mathbf{R}(t)$ is a matrix determined by a rotary positional embedding. After cross-attention, we apply a task-specific linear layer $W_k : \mathbb{R}^D \to \mathbb{R}^{D_k}$ where $D$ is the dimension of each output tokens $\mathbf{y}_{\ell k i}$, and $D_k$ is the dimension of the target for task $k$. For regression tasks, $D_k$ is the dimensionality of the variable being decoded, for example, $D_k = 1$ for running speed and $D_k = 2$ for pupil location. For classification and segmentation tasks, $D_k$ will be the number of classes.

The entire network is trained end-to-end on all tasks simultaneously. We use mean-squared error loss for regression tasks, and cross-entropy loss for classification and segmentation tasks. The losses are summed and weighted based on the number of predictions for each task (i.e. the more predictions requested the lower the weight for any specific token).

With this set-up, we are able to decode multiple labels simultaneously during both training and inference. For example, in a context window of one second where the stimulus presentation is drifting gratings, we can decode on four different tasks: running speed regression, pupil location regression, orientation classification, and temporal frequency classification. Because running speed and pupil diameter are monitored at a frequency of 30 Hz we will have 30 outputs for each of these tasks over a 1 s context window. For the other two tasks, we will have only one output, since these values are constant throughout the context for drifting gratings. Thus, we will have a total of 62 query tokens and outputs. But, notably, our multi-modal decoder is able to handle any number of query tokens, from different tasks due to its flexible design.

# 3 RESULTS

## 3.1 TRAINING A MODEL ON DIVERSE MIXTURES OF NEURAL RECORDINGS

**Datasets.** To train our model, we leverage the Allen Brain Observatory (BO) calcium dataset (de Vries et al., 2020). The Allen BO is the largest collection of *in vivo* two-photon calcium imaging data, acquired from the visual cortex of awake mice. It contains neuronal activity in six different cortical areas: Primary Visual Area (VISp), Posteromedial visual area (VISpm), Anteromedial visual area (VISam), Rostrolateral visual area (VISrl), Anteromedial visual area (VISal), Lateral visual area (VISl), in response to various visual stimuli, including both simple drifting gratings (eight different orientations), static gratings, natural scenes, and naturalistic movies (Appendix B). Each experiment consists of three one-hour long imaging sessions, one for each mouse, imag-

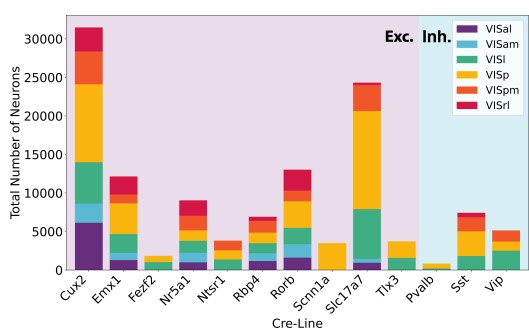

Figure 2: **Data diversity in the Allen Brain Observatory.** A breakdown of the number of neurons for each Cre-line by brain region. The Cre-lines are organized by Excitatory and Inhibitory lines from left to right, and the visual areas are depicted with different colors.

ing depth, and visual area, recorded at various depths, ranging from 150 to 600 μm from the surface of the brain, each of which can be classified into different layers (L2/3, L4, L5, or L6).

The mice expressed a genetically encoded calcium sensor (GCaMP6f) in specific genetic sub-types of neurons using Cre driver lines. There were two pan-excitatory lines where the calcium sensor was expressed in almost all excitatory neurons (Emx1, Slc17a7), eight lines where it was expressed only in specific sub-types of excitatory neurons (Cux2, Rorb, Scnn1a, Nr5a1, Rbp4, Fezf2, Tlx3, Ntsr1) and three lines where it was expressed only in specific sub-types of inhibitory neurons (Vip, SST, PV). Each of these cellular sub-types show distinct response properties (de Vries et al., 2020), and very different densities of neurons in different brain regions (Figure 2). Our challenge here was to see if we could incorporate data from these diverse sub-types to improve decoding.

**Experimental setup.** Each recording session is divided into training, validation and test splits with a 60/10/30 ratio. Given the unstructured nature of the recording, we tried to maintain these ratios across each stimulus epoch. When reporting accuracies, we perform evaluation on the test split of each session, and report the average across sessions. Note that there is high variability in the per-session decoding accuracy, likely because we are working with heterogeneous data, with half of the sessions having less than 62 neurons and some as few as 5. We note that in total we use 1335 3-hour sessions for training our model. During training, we leverage neuron dropout where a random subset of the population is removed from the context window.

For the purposes of transfer, we holdout sessions from 30 animals that are never seen during training. We use the training split of each heldout session to finetune POYO+ , then report the test accuracy. We use gradual unfreezing where we first unfreeze the unit and session embeddings then the multi-modal decoder then the encoder. We find that in practice, the model already converges with just the unit and session embeddings unfrozen. More details can be found in Appendix C.2.

**The benefits of multi-session training.** Given the inherent diversity of our dataset, we first investigated the scaling properties of the model when trained on a specific context. For this experiment, we used the drifting gratings stimulus condition and trained a variant of POYO that we designed to work with our new tokenization strategy for regular time series data as input. The model was trained on all sessions that include drifting gratings, we call this model POYO (DG) (DG for Drifting Gratings). For comparison, we trained two groups of single-session baseline models—(i) MLP and (ii) TCN—on each of the 403 sessions collected under the DG condition. Each model was extensively tuned using a validation set, we compared a total of more than 80,000 models that were trained during hyperparameter tuning. When compared to these single-session baselines, we observed significant improvements on test trials, with the multi-session model providing nearly a 10% performance gain on average relative to the MLP baseline (Figure 3A). We also compared POYO+ to a contrastive multi-session approach for training that uses labeled data to create positive views across

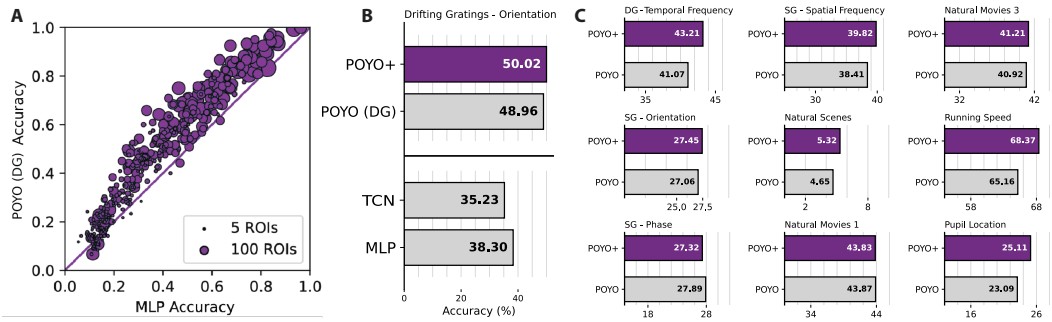

Figure 3: **The benefits of a multi-session, multi-task neural decoder.** A. POYO (DG) compared to an MLP on in-distribution datasets. B. POYO+ and POYO (DG) compared to MLP and TCN baselines on the drifting gratings orientation prediction task. C. Comparing task-wise variants of POYO to POYO+ . POYO+ is a single model trained on 12 tasks simultaneously, while we have 12 variants of POYO.

animals (CEBRA, (Schneider et al., 2023b), see Appendix D.2 for more details). We found that our approach again provides superior decoding performance for multi-session data. In conclusion, we find that our single-task variant of POYO consistently beats different baselines (Figure 3B). This highlights the advantages of aggregating information across multiple sessions for this decoding task.

**Incorporating multi-task learning across diverse contexts.** After confirming the benefits of multi-session training in the drifting gratings condition, we extended our model to include additional stimulus contexts as well as behavioral decoding tasks, such as pupil diameter and running speed. In Figure 3C, we compare POYO+ to the single-task POYO variants each trained on one of the 12 behavioral and contextual decoding tasks. We find that incorporating multiple tasks into the output space of the decoder improves performance on a number of tasks, especially the behavioral decoding tasks. We hypothesize that multi-task training provides two key benefits. First, the inclusion of dense prediction tasks (e.g., running speed and pupil location) provides auxiliary information to the decoder, improving overall predictions. Second, it engages a broader population of neurons in the model, as some neurons responded more strongly to behavioral changes than to specific stimulus contexts. This suggests that multi-task learning not only enhances decoding performance but also better captures the rich, context-dependent modulation of neural activity.

## 3.2 TRANSFER ACROSS BRAIN REGIONS AND CRE-LINES

**Across-region transfer.** The visual cortex exhibits distinct activity patterns across its regions, raising the question of whether models trained in one area can transfer effectively to others. Our results show that many regions provide a solid basis for transfer, with VISal and VISl delivering the strongest overall performance. Specifically, the POYO+ model trained on VISal achieved the highest accuracy in VISal itself (73.84%) and demonstrated good transfer to VISam (52.19%), VISl (59.78%), and VISp (52.14%). Similarly, the POYO+ model trained on VISl also performed well, achieving high transfer accuracy in VISal (72.22%) and VISpm (35.34%). Interestingly, VISp, a primary visual area, exhibited strong transfer to VISal (74.01%) and VISl (58.74%), suggesting its foundational role in cross-region transfer.

Table 1: **Transfer across held-out brain areas.** The results from training on one region and finetuning on new held-out sessions in another region, are compared with POYO+ trained on all brain areas. All of the across-line results that perform better than the base within-line model are highlighted in green. The best performing model for each testing condition are in boldface and the number of sessions in each test condition are in parentheses.

| Model | Training sessions | Average (30) | VISal (4) | VISam (2) | VISl (8) | VISp (13) | VISpm (3) |
|---|---|---|---|---|---|---|---|
| MLP | - | 43.36% | 56.39% | 35.68% | 45.65% | 42.17% | 30.13% |
| POYO+ (VISal) | 119 | 54.12% | 73.84% | 52.19% | 59.78% | 52.14% | **36.37%** |
| POYO+ (VISam) | 112 | 47.00% | 63.20% | 47.54% | 49.58% | 45.67% | 28.15% |
| POYO+ (VISl) | 310 | 53.98% | 72.22% | 51.17% | 58.01% | 52.52% | 35.34% |
| POYO+ (VISp) | 419 | 53.67% | **74.01%** | 48.89% | 58.74% | 52.15% | 31.73% |
| POYO+ (VISpm) | 258 | 51.94% | 69.10% | **51.83%** | 56.88% | 50.72% | 30.41% |
| POYO+ (All) | 1335 | 55.96% | 73.96% | 49.39% | **61.46%** | **54.84%** | 36.01% |

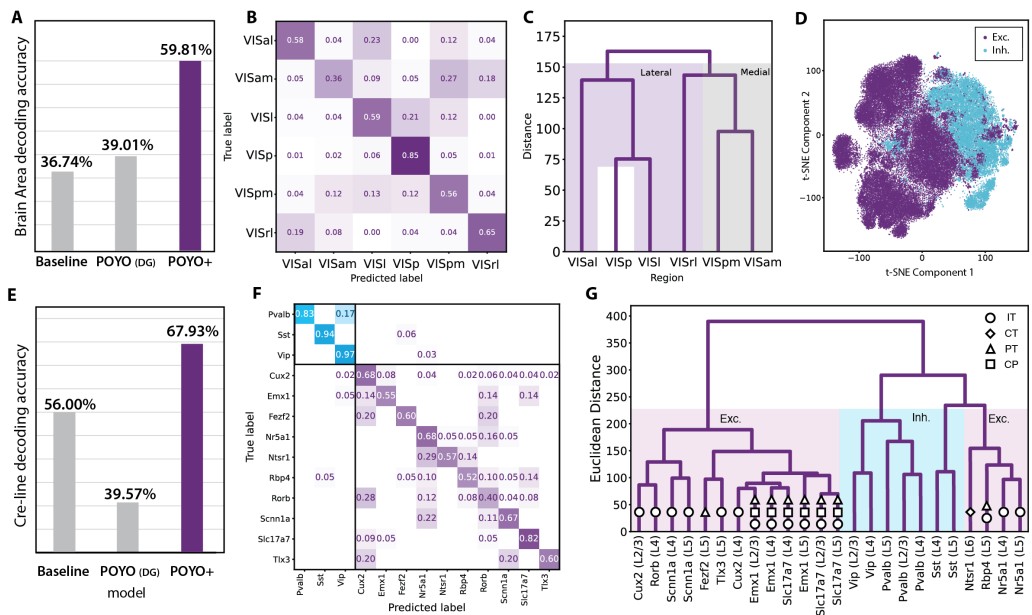

Figure 4: **Examining the organization of the latent embeddings by Cre-line and brain region.** A. Balanced accuracy for brain area classifcation based on hand-crafted features versus session-averaged latents from POYO+ B. Confusion matrix for brain area classification C. Dendrogram of average latents by brain area D. t-SNE projection of POYO+ embeddings. E. Balanced accuracy for Cre-line classification based on hand-crafted features versus session-averaged latents from POYO+ F. Confusion matrix for Cre-line classification G. Dendrogram of average latents by Cre-line.

These results highlight that training on higher-order areas like VISal and VISl facilitates superior transfer to other regions, likely due to their integrative roles in visual processing. Furthermore, the model trained on all regions, POYO+ (All), achieved the highest overall performance (55.96%), demonstrating that exposure to data from multiple brain areas improves generalization across unseen regions. The finding that transfer performance improves with more diverse training data emphasizes the importance of using data from different brain areas and animals, which enables the model to capture a wider range of neural dynamics and improves its adaptability across regions.

**Across-Cre-line transfer.** Next, we evaluated whether our models could transfer effectively across excitatory and inhibitory Cre-lines. While our model trained on all lines surpasses the other Cre-line specific models, we find that the model trained on inhibitory Cre-lines does surprisingly well on the drifting gratings task. Specifically, the model trained on inhibitory data, POYO+ (Inh), achieved the best performance on inhibitory data and also surpasses the model trained on excitatory data alone.

Table 2: **Transfer across held-out Cre-lines.** The decoding accuracy for the drifting gratings task for held-out animals for models trained on Excitatory lines only, Inhibitory lines only, and All lines. The performance is measured on (16) Excitatory sessions and (8) Inhibitory sessions spanning different lines. All of the across-region results that perform better than the base within-region model (purple) are highlighted in green. The best performing model for each testing condition is in boldface.

| Model | Training sessions | Average (30) | Excitatory (16) | Inhibitory (8) |
|---|---|---|---|---|
| MLP (Base) | - | 43.36% | 48.47% | 26.61% |
| POYO+ (Exc) | 987 | 54.74% | 55.40% | 30.01% |
| POYO+ (Inh) | 348 | 55.85% | 62.89% | **31.96%** |
| POYO+ (All) | 1335 | **55.96%** | **62.93%** | 31.80% |

## 3.3 ANALYSIS OF LATENT EMBEDDINGS

Although our model has been trained to decode the stimulus information or behavior, we wanted to understand the extent to which the model has learned session-level differences that capture biologically meaningful information about the Cre-line and brain regions for each session. To test this, we examined the latents at the output of the encoder in POYO+ . To obtain the session-level latent

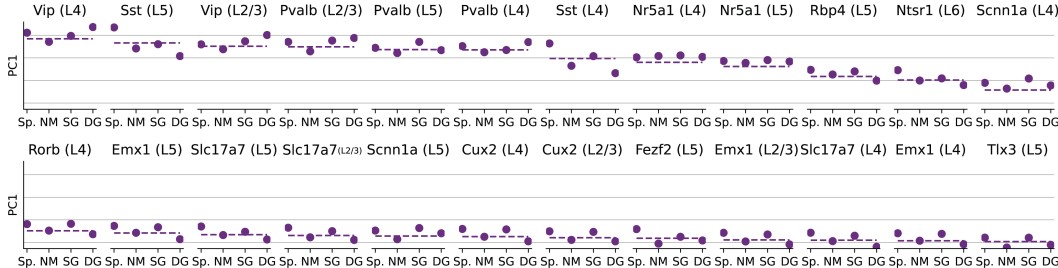

Figure 5: **Deviation of Cre-line averaged embeddings from baseline when considering one visual stimulus type at a time.** We use 1-dimensional PCA projection and plot the average latent embeddings for the Spontaneous (Sp), Natural Movies (NM), Static Gratings (SG), and Drifting Gratings (DG) for each Cre-line separated by layer (L2/3, L4, L5, L6).

embeddings, we average the latents across randomly sampled 1 s context windows in the recording. See Appendix E for further details on this analysis.

**Latent representations of distinct brain regions.** First, we examined the latent embeddings for sessions from different brain regions. Using a linear decoder trained on the latent embeddings for different sessions, we were able to decode the brain region of the recording session with high accuracy (Figure 4A), much higher than using standard POYO latents or a baseline using hand-crafted features used for cell-type analysis (see Appendix E.2 for more details). Examination of the confusion matrix for decoding from the POYO+ latents showed that some regions were more likely to be confused with each other by the decoder, for example VISal was likely to be confused for VISl (Figure 4B). This suggests that the underlying latents were more or less similar for different groups of brain regions. To examine this, we conducted hierarchical clustering on the latents for sessions from different brain regions. Interestingly, we found that the hierarchical clustering distinguished medial regions clearly from lateral ones, and the lateral region which was most closely clustered to the medial regions (VISrl) is in fact the most medial of the lateral regions (Figure 4C). This demonstrates that the learned latent embeddings reflect the actual anatomy of visual cortex, despite the model having received no information about this anatomy.

**Latent representations of cellular sub-types.** We then conducted similar analyses on the sessions for different cellular sub-types. First, we examined the projection of the latents to a lower-dimensional space, using t-SNE. Interestingly, we observed clear separability for session latents from inhibitory and excitatory sub-types (Figure 4D), suggesting that the latents had learned to capture cellular sub-type distinctions. Next, to quantify the separability of different Cre-lines we again trained a linear decoder on top of the extracted latents, this time to decode the Cre-lines. We find that the organization of Cre-line is quite impressive, with a overall 67.93% accuracy in decoding, compared to 56% and 39.57% for a transcriptomic baseline (Schneider et al., 2023a) and standard POYO, respectively (Figure 4E). These results suggest that to really understand the neural activity across different cellular sub-types it is crucial to consider a multi-task view.

We also then examined the confusion matrix for the Cre-lines. As with brain regions, we observed that some cellular sub-types are more or less likely to be confused with each other, e.g. excitatory neurons are far more likely to be confused with other excitatory neurons than inhibitory neurons (Figure 4F). To understand the structure in the latents better, we again conducted hierarchical clusering for the latents from different sessions based on the Cre-lines (Figure 4G). We observed several interesting results. First, as expected given the low-dimensional embeddings, the inhibitory neurons were more closely grouped together, and the clusters within the inhibitory types reflected true biological distinctions (e.g. the different types of Pvalb sessions were clustered together). Second, the clustering had some relationship to the laminar position of the cellular sub-types, such that cells in the supragranular layers (L2/3) tended to cluster with each other or sub-types from L4. Finally, we observed that the clusters appeared to respect the axonal projections of the different cellular sub-types. For example, intratelencephalic (IT) neurons tended to be closer to each other in the hierarchy than to pyramidal tract (PT) neurons. The fact that the latent embeddings for the different sessions reflected these real biological differences in physiology and anatomy was surprising but encouraging, and suggests that our model can be used to reveal meaningful biological relationships.

**Visualization of task-dependent modulation of embeddings.** Finally, in examining the latent embeddings for the different sessions we wanted to see whether they reflected stimulus-specific distinctions between cellular sub-types. In Figure 5, we aggregate the latent embeddings within each stimulus context, and examine them across Cre-lines. We compute the first principal component (PC) and project the latents from each task onto the PC, and report the multi-task aggregated embedding as the dotted baseline. We are able to see the main source of variation in the latents and how it maps onto the different stimulus contexts and Cre-lines.

First, we note outliers like Sst, an interneuron sub-type that behaves similarly to other interneurons (Vip, Pvalb) during all contexts except for the drifting gratings context, where it deviates from the baselines in the opposite direction from other interneurons. This makes sense based upon prior work which shows that VIP are suppressed during drifting gratings (de Vries et al., 2020). We also observe subtle differences across different layers in the same Cre-line, like Pvalb neurons in layer 5 maintaining their baseline dynamics during drifting gratings, while we observe a shift in Pvalb neurons in L 2/3 and 4. In excitatory neurons, we see overall similar modulation across stimuli, but an interesting stability in Nr5a1 neurons which appear to be consistent regardless of the stimulus. These data demonstrate another advantage of a multi-task model trained on heterogeneous data: it allows us to compare distinct cellular sub-types in a stimulus/context dependent manner.

## 3.4 TRANSFER TO OPENSCOPE DATASET

In this experiment, we tested whether we could transfer from our pretrained model to another open dataset with different recording conditions and stimuli, collected as part of The Allen Institute Mindscope's OpenScope program (Gillon et al., 2023) (Table 3). This dataset consists of responses of neurons in VisP from two Cre driver lines targeting the soma and dendrites of distinct excitatory populations (Cux2, somatic; Rbp4, dendritic). When we finetune our pretrained multi-task model on sessions from this new

Table 3: **Transfer from POYO+ to a new Openscope dataset.** We compare the decoding performance on a 3-class task for an MLP, single task POYO (DG) and POYO+ , both finetuned for the new task.

|  | Avg (5) | Dend (3) | Soma (2) |
|---|---|---|---|
| MLP | 72.94% | 73.51% | 72.10% |
| POYO (DG) | 79.70% | 81.74% | 76.76% |
| POYO+ | **81.91%** | **84.19%** | **78.49%** |

dataset, we find a near 9% improvement with POYO+ over the MLP baseline, and a 2% improvement over the DG-only pretrained model. If we break these numbers down by the type of recording, we find that the improvements on Dendritic sessions benefit the most, with a 10% improvement with POYO+ despite the fact that we did not pretrain on dendritic data which varies considerably from somatic data (see Appendix D.1).

## 3.5 TRANSFER TO A HIPPOCAMPUS DATASET

To understand how our pretrained model can transfer to non-visual areas, we obtained a calcium imaging dataset from the hippocampus as mice explored differently shaped $75 \times 75$ cm open mazes Lee et al. (2025) (10 sessions). We used this data to decode the animal's position. We compared POYO+ pretrained on the Allen Observatory data and one trained from scratch. Importantly, for the pretrained model, we froze all weights except for the cross-attention layers at the input and output.

Table 4: **Transfer from POYO+ trained on visual areas to data recorded in hippocampus.** We finetune POYO+ to perdict the position of the mouse in the maze. We report the mean euclidean error.

|  | Error |
|---|---|
| POYO+ from scratch | 10.33 cm |
| POYO+ pre-trained | 9.58 cm |

As we now show in Table 4, the model pretrained on the visual data performs better than the model trained from scratch. Moreover, both models are better in accuracy than the Bayesian decoding model applied by Lee et al. (2025), which produced a position decoding error of 12 cm. This confirms that our model is capable of learning representations that transfer from data collected in one region to another, even if those regions are functionally very distinct.

## 4 RELATED WORK

**Multi-session training for neural recordings.** Neural decoding has been extremely challenging, especially when trying to transfer to recordings from different days or animals. This is due to a number of issues, one of the most basic being the fact that each recording session has a different set

of neurons associated with it Dabagia et al. (2022). Recently, there has been progress in demonstrating that large transformer architectures can be trained on multiple recording sessions to successfully decode and transfer to new brains and tasks (Azabou et al., 2024; Schneider et al., 2023b; Ye et al., 2024). In particular, the POYO architecture and tokenization scheme introduced by (Azabou et al., 2024) provided a scalable solution to this problem by using a latent embedding scheme for encoding individual neural dynamics, along with a PerceiverIO encoder-decoder architecture to efficiently decode high-resolution behavioral outputs from brain signals. However, existing approaches for multi-session training (Ye et al., 2024; Azabou et al., 2024) have still only been applied to sessions containing the same brain regions and neural cell types, and for the same decoding tasks. Here, we investigate whether it is possible to train this type of transformer architecture on increasingly heterogeneous datasets from different areas of the brain and genetically distinct cell types. Additionally, previous methods have been applied to tens to a hundred sessions, POYO+ moves to the next order of magnitude with 1,335 sessions.

**Deep learning methods for analysis of neural population activity in calcium data.** As calcium imaging datasets are improving, there have been a number of works that have focused on building neural population models for calcium data analysis (Schneider et al., 2023b; Prince et al., 2021; Zhu et al., 2022; de Oliveira Fonseca et al., 2023; Park et al., 2023). Many of these methods are designed for dynamical systems analysis and forecasting and thus the decoding accuracy is not the focus. Most similar to ours is CEBRA (Schneider et al., 2023b) as the method provides a way to incorporate supervised data for training a decoder. CEBRA uses behavioral labels to define positive views to align labeled examples. This method requires either trial-aligned data to concatenate many animals into a pseudopopulation or multi-data stitching approaches where you need to train a different encoder head for each dataset. In contrast to these methods, we demonstrate flexible decoding of multiple behavior modalities and contexts. Additionally, due to the rich encoder and tokenization method, we also show that our method can be used to decipher different cell types and regions from the embeddings of population-level neural activity.

## 5 CONCLUSION

In this work, we introduced a novel multi-session, multi-task neural decoding approach and applied it to a large-scale dataset from the Allen Institute for Brain Science. Our results demonstrate that scaling up to incorporate multiple tasks can effectively extract meaningful insights into neural circuit diversity, relying solely on neural activity without prior knowledge of neuron sub-types or brain regions. The latent representations learned by our model revealed clear distinctions between cellular sub-types and brain regions, underscoring the power of multi-task learning to uncover biological structures embedded within neural data.

Our approach further highlights the value of integrating diverse neural datasets, offering a powerful framework for comparing and contrasting activity patterns across different cell populations, brain regions, experimental conditions, and even across animals. This comparative analysis enables us to observe how distinct neural populations interact and coordinate their activity, yielding insights that would be difficult to capture through isolated experiments. By leveraging a large collection of diverse recordings, we were able to learn patterns of neural population dynamics that generalize across subjects and conditions.

In developing this approach, we utilized the largest and most diverse dataset publicly available, covering multiple brain regions, Cre-lines, and experimental sessions. This comprehensive dataset allowed us to train models that generalize across varying conditions. However, despite the dataset's richness, we have only tested our model in visual cortex and the current work does not incorporate other regions or species, which could further validate the generalizability of our model. Additionally, our model does not yet incorporate neuron activity prediction, which could help to further improve the model's ability to understand the dynamics of different cellular sub-types and regions. Addressing these limitations by expanding to additional brain areas, species, and neuron prediction tasks will be crucial areas in future research.

Looking forward, our framework has the potential to go beyond traditional limitations in neuroscience by algorithmically stitching together neural populations from different experiments. Applying this model to even larger datasets, across species and brain regions, could lead to discoveries in how neural circuits are organized and how they change over learning and disease.

ACKNOWLEDGEMENTS

We would like to thank J. Quinn Lee and Mark P. Brandon for sharing the Lee et al. (2025) dataset and providing feedback on the experiment. We would also like to thank Saskia de Vries for providing feedback on the analysis. BAR acknowledges support from NSERC (Discovery Grant: RGPIN-2020-05105; Discovery Accelerator Supplement: RGPAS-2020-00031; Arthur B. McDonald Fellowship: 566355-2022), CIFAR (Canada AI Chair; Learning in Machine and Brains Fellowship), and the Canada Excellence Research Chairs Program. KXP was supported by a PhD Fellowship from Google DeepMind. ED acknowledges support from NIH award 1R01EB029852-01, NSF award IIS-2146072 as well as generous gifts from the CIFAR Azrieli Global Scholars Program. MA acknowledges support from the National Science Foundation and by DoD OUSD (R&E) under Cooperative Agreement PHY-2229929 (The NSF AI Institute for Artificial and Natural Intelligence).

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

# APPENDIX

## A   WHY BUILD A MODEL ACROSS DIVERSE CELL TYPES AND BRAIN REGIONS?

### A.1   CHALLENGES

Sensory processing occurs across multiple interconnected brain areas, which collectively transform their inputs into diverse, hierarchically organized representations. Consequently, the representation at the population level in different brain regions exhibit significant variation, particularly as we move up the hierarchy (Tong et al., 2023; Esfahany et al., 2018). For example, when we examine the representations in primary visual cortex (VisP) and compare them to those in a higher-order visual area like rostro-lateral visual cortex (VisRL), the neurons appear to respond to different pieces of information, e.g. VisP neurons show strong orientation tuning while VisRL neurons do not (Esfahany et al., 2018). This variability highlights the complexity of neural "conversations" in the brain, as neurons in different areas encode information in unique, complementary ways that are essential for comprehensive sensory processing.

In addition to understanding the variability across brain regions, understanding the information processing properties of different genetically defined cellular sub-types is crucial. Luckily, modern neuroscience tools allow researchers to target their recordings to distinct cellular sub-types using Cre-lox technology (Li & Yang, 2023). As a result, it is now known that different cellular sub-types exhibit unique response patterns to the same stimuli (Zeng & Sanes, 2017). For example, different sub-types in visual cortex are distinctly modulated by locomotion differently, and show distinct response profiles to the same sensorimotor conditions (Dipoppa et al., 2018). As well, some cellular sub-types are more numerous than others, with very high variance in the sparsity/density of recordings from different Cre-lines (de Vries et al., 2020). Training a decoding model across these distinct cellular sub-types is challenging due to the heterogeneous and context-dependent nature of their responses, but it is also critical for the ability to train across highly diverse neural recordings.

### A.2   OPPORTUNITIES

What could we achieve with a neural decoding model that combines information across brain regions, cell types, and contexts? First, such a model could serve as a generalist framework—pretrained for any brain region or Cre-line—allowing for fine-tuning on new datasets regardless of the regions or sub-types involved. By stitching together data from different experiments, we could algorithmically link neural populations that are otherwise studied in isolation. For example, in a single mouse or experiment, we typically record from only one sub-type of cell in a specific brain region. But by integrating multiple datasets, we could begin to understand how different cellular sub-types work together across regions in a unified system, gaining a broader view of neural dynamics than would be possible using data from a single experiment.

Combining diverse neural datasets offers a powerful opportunity to compare and contrast activity patterns across different cell populations, brain regions, experimental conditions, and even across animals. This comparative approach allows us to observe how distinct neural populations interact and coordinate their activity in ways that individual experiments cannot capture. By integrating diverse datasets, we can explore how neural circuits change during learning, identifying patterns of adaptation that generalize across subjects. Similarly, we can track how neural activity evolves in the context of disease, revealing key differences between healthy and pathological states. Altogether, by training generalist models on a wealth of diverse data, we can uncover neural circuit interactions that remain hidden when analyzing data from a single source.

## B   DETAILS ON THE ALLEN BRAIN OBSERVATORY DATASET

Below we provide a breakdown of all the experiments in the Allen Brain Observatory (Table A1) where we see the number of sessions per brain region and the different tasks.

| TASK | TARGETED STRUCTURE | | | | | | |
|------|------|----|----|---|---|----|----|
|  | **All** | AL | AM | L | P | PM | RL |
| Drifting (DG) | **456** | 41 | 38 | 106 | 144 | 87 | 40 |
| Static (SG) | **456** | 41 | 38 | 106 | 144 | 87 | 40 |
| Scenes (NS) | **456** | 41 | 38 | 106 | 144 | 87 | 40 |
| Movies (NM) | **1368** | 123 | 114 | 318 | 432 | 261 | 120 |
| Sparse Noise (LSN) | **456** | 41 | 38 | 106 | 144 | 87 | 40 |

Table A1: Session breakdown

## C  MODEL DETAILS AND EXPERIMENTAL SETUP

### C.1  ROTARY POSITION EMBEDDINGS (RoPE)

Position of tokens in our model are determined by their timestamps. Thus, we allow the positions to be continuous values, which is different from the conventional application of RoPE in domains such as language and vision. For this, we define a custom rotation matrix $\mathbf{R}(t)$ as follows:

$$\mathbf{R}(t) = \begin{bmatrix} \mathbf{R}_{2\times2}(t, T_1) & 0 & \cdots & 0 \\ 0 & \mathbf{R}_{2\times2}(t, T_2) & \cdots & 0 \\ \vdots & \vdots & \ddots & \vdots \\ 0 & 0 & \cdots & \mathbf{R}_{2\times2}(t, T_{D/2}) \end{bmatrix}$$

With each $2 \times 2$ sub-matrix being defined as:

$$\mathbf{R}_{2\times2}(t, T) = \begin{bmatrix} \cos(2\pi t/T) & -\sin(2\pi t/T) \\ \sin(2\pi t/T) & \cos(2\pi t/T) \end{bmatrix}$$

Here, $T_1, T_2, \ldots, T_{D/2}$ denote the "time-period" of different sinusoids. These decide what resolution and range the model is capable of resolving. We set these time-periods to be logarithmically spaced from $0.1ms$ to $4.0s$.

### C.2  TRAINING DETAILS

We use the LAMB optimizer You et al. (2019) with weight decay to train POYO+ . The learning rate is held constant, then decayed towards the end of training (last 25% of epochs), using a cosine decay schedule. We train POYO+ for 300 epochs using 8 Nvidia H100s with a global batch size of 4800.

POYO+ is trained with 8 transformer blocks: 2 cross-attention blocks at the input and output and 6 self attention blocks. We use 128 latent tokens with a dimension of 128. We use multi-head attention with cross attention layers using 2 heads, and self-attention layers using 8 heads. We use dropout when computing attention, and in the feed-forward networks (dropout rate 0.2).

POYO+ is trained on a total of 12 tasks, which are detailed in Table A2. The weight for each task is determined based on the scale of the loss from an initial model. The regression targets are z-scored based on the mean and standard deviation computed from the training trials across all training sessions.

### C.3  MULTI-MODAL DECODER IN POYO+

In Figure A1, we detail the architecture of the multi-modal decoder used in POYO+ . All tokens regardless of the task are processed through the same cross-attention layer, and a custom router groups together tokens from the same task and feeds them to a task-specific linear decoder.

Table A2: Summary of tasks, dimensions, loss functions, and reported metrics used in POYO+ .

| Task | Dim | Type | Loss | Weight | Metric |
|---|---|---|---|---|---|
| Drifting gratings orientation | 8 | Classification | bce | 1.0 | Accuracy |
| Drifting gratings temporal frequency | 5 | Classification | bce | 1.0 | Accuracy |
| Natural movie one frame | 900 | Segmentation | bce | 0.25 | Acc within 1s |
| Natural movie two frame | 900 | Segmentation | bce | 0.2 | Acc within 1s |
| Natural movie three frame | 3600 | Segmentation | bce | 0.2 | Acc within 1s |
| Locally sparse noise frame | 8000 | Segmentation | bce | 1.0 | Accuracy |
| Static gratings orientation | 6 | Segmentation | bce | 1.0 | Accuracy |
| Static gratings spatial frequency | 5 | Segmentation | bce | 1.0 | Accuracy |
| Static gratings phase | 5 | Segmentation | bce | 1.0 | Accuracy |
| Natural scenes | 119 | Segmentation | bce | 0.3 | Accuracy |
| Running speed | 1 | Regression | mse | 1.5 | $r^2$ |
| Pupil location | 2 | Regression | mse | 8.0 | $r^2$ |

# D   ADDITIONAL EXPERIMENTS

## D.1   TRANSFER EXPERIMENTS TO OPENSCOPE DATASET

In another set of experiments, we examined our ability to transfer from our pretrained model to another open dataset with different recording conditions and stimuli, collected as part of The Allen Institute Mindscope's OpenScope program (Gillon et al., 2023). This dataset consists of responses of neurons in VisP from two Cre driver lines targeting distinct excitatory populations (Cux2 and Rbp4) recorded at two different depths. The two different depths were used to target either the cell bodies or distal apical dendrites of the neurons. Thus, these recordings contained responses from dendritic processes (Figure A2 in the Appendix), not just neural cell-bodies, which allowed us to examine our ability to transfer to dendrites.

We tested our ability to finetune our pretrained model to decode the orientation of Gabor stimuli, from either the activity of cell bodies (Soma) or dendrites (Dend) (Table 3). Our results suggest that there is indeed a benefit in pretraining on another large dataset. When we compare to the MLP baseline, we find a near 7% improvement in performance with POYO (DG) and 9% improvement with POYO+ which is pretrained on much more diverse tasks. When we break these numbers down by the type of recording, we find that the improvements on Dendritic sessions benefit the most, with

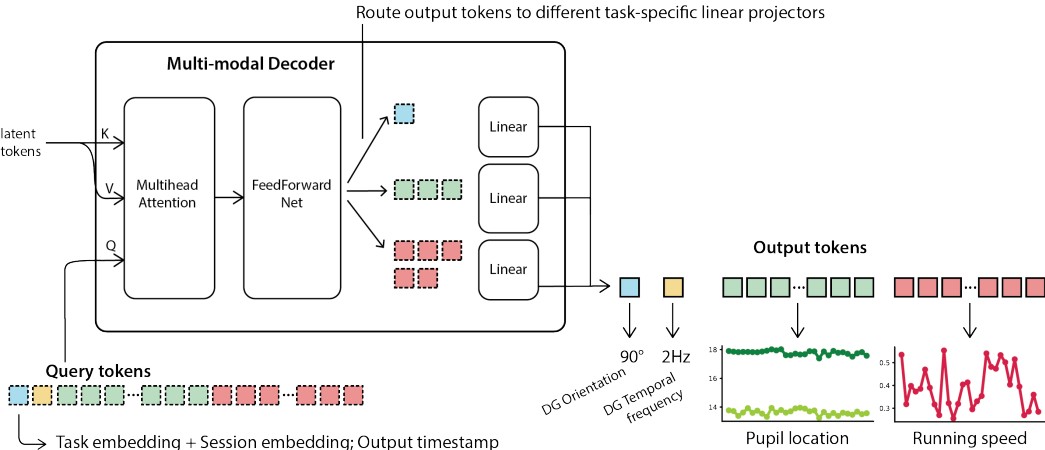

Figure A1: Detailed architecture of the decoder in POYO+ . All tokens from all tasks are processed together throught the same cross-attention layer. A router is used to group together tokens from the same task, for which a task-specific linear decoder is used to project the token into the target dimension.

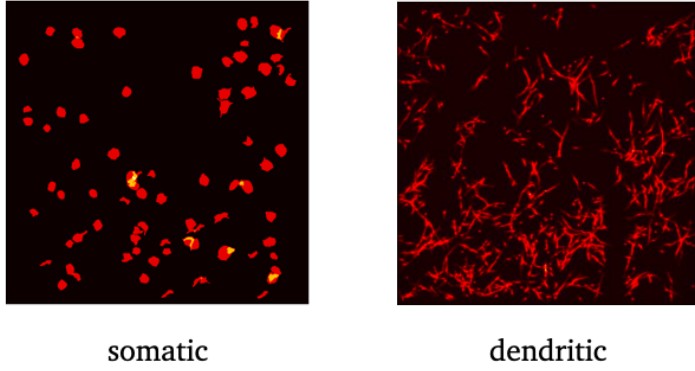

somatic                                    dendritic

Figure A2:    Visualizations of somatic and dendritic region-of-interests.

over 10% improvement with POYO+ despite the fact that we did not pretrain on dendritic data which varies considerably from somatic data.

Table A3: **Transfer from POYO+ to a new Openscope dataset.** Here, we compare the decoding performance on a 3-class task for a MLP baseline, the single task POYO (DG) baseline finetuned for the new task, and POYO+ finetuned.

|            | Avg (5)  | Dend (3) | Soma (2) |
|------------|----------|----------|----------|
| MLP        | 72.94%   | 73.51%   | 72.10%   |
| POYO (DG)  | 79.70%   | 81.74%   | 76.76%   |
| POYO+      | **81.91%** | **84.19%** | **78.49%** |

## D.2    COMPARISON OF CEBRA AND POYO+

We compare POYO+ to CEBRA (Schneider et al., 2023b), which is a state-of-the-art neural latent model that supports multi-session training. CEBRA is a self-supervised method that uses auxiliary context variables to form representations that are guided by the behavior or stimulus labels. We test CEBRA in two different settings. In first setting, we replicate the drifting gratings experiment in our paper. In the second, we replicate the experimental setup in Schneider et al. (2023b) using the pseudomouse method.

### D.2.1    DRIFTING GRATINGS TASK

We pretrain multi-session CEBRA on the drifting gratings data from all sessions in the Allen Brain Observatory. We then fit a linear decoder on top of the representations learned by CEBRA. We found that this achieves a performance of 18.66%. For comparaison, the MLP baselines averaged across the same session reach a performance of 38.30%. POYO (DG) which is trained on the same data as CEBRA, in a multi-session fashion, reaches a performance of 48.96%.

### D.2.2    NATURAL MOVIE FRAME PREDICTION TASK USING PSEUDOMOUSE

In a second attempt at comparing CEBRA and POYO+ , we replicate the CEBRA pseudomouse experiment using the same Allen Brain Observatory dataset natural movie subset used in the original CEBRA paper Schneider et al. (2023b).

Following CEBRA's experimental setup, we excluded sessions recorded from mice expressing three inhibitory Cre-lines and Ntsr1-Cre_GN220. Instead of using the traces provided by the Allen Institute, we also used the preprocessed and aligned version of the traces from Deitch et al. (2021) (https://github.com/zivlab/visual_drift/tree/main/data). This dataset provides traces aligned with movie frames for natural movie one stimulus, facilitating aggregation of neurons across animals. We note that we only compared CEBRA's VISp results here because the main findings and CEBRA's best performance were reported from VISp out of the six brain regions.

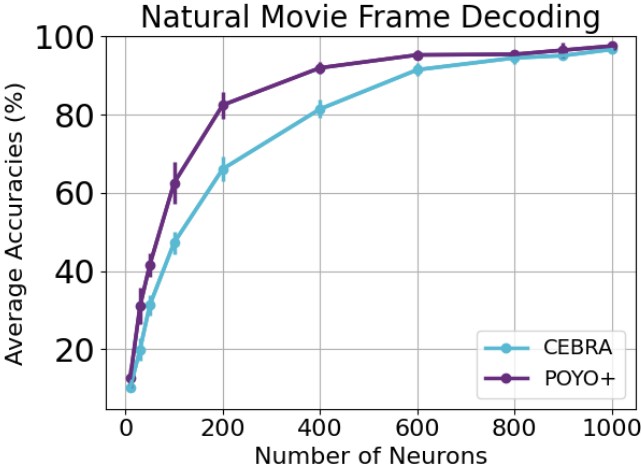

Figure A3:  **Decoding accuracy comparison between POYO+ and CEBRA through neuron concatenation.** POYO+ consistently outperforms CEBRA across all pseudomouse sizes on a natural movie frame decoding task.

We used all sessions from VISp available in the dataset . In each session, the natural movie one video was played 10 times consecutively. We split the training set in the exact same way as CEBRA, using the first eight repeats as the training set, the ninth as the validation and the tenth as the test set.

We constructed a "pseudomouse" by first aggregating all neurons from all subjects, only including the neurons that were tracked across all three types of sessions, ensuring we have the same neuron pool as CEBRA. Since we could not identify the exact same neurons used in the CEBRA experiments, we employed the same neuron pool and method, using a random sampler to sample different numbers of neurons. This allowed us to construct pseudomice of ten different population sizes: {10, 30, 50, 100, 200, 400, 600, 800, 900, 1000}. For each size, we sampled randomly with five different seeds to ensure reproducibility, generating 50 pseudosubjects in total.

We tested the model's frame decoding performance with a 900-class classification task. Since the movie frames are very similar within a small time window, we followed CEBRA's approach: if the prediction was within a one-second window, we counted it as a correct prediction.

We train the model for 3000 epochs for pseudomice of sizes {10, 30, 50, 100, 200, 400} neurons, 1500 epochs for sizes {600, 800} neurons and 1000 epochs for sizes {900,1000} neurons. We then computed the averages of decoding accuracy for each size, as well as the standard error of the mean (s.e.m.). We report CEBRA results from their repository for reproducing figures (https://github.com/AdaptiveMotorControlLab/cebra-figures).

The results show that POYO+ outperformed CEBRA for pseudosubjects of all sizes (Figure A3). Notably, POYO+ outperformed CEBRA by a large margin for pseudosubjects containing fewer neurons. For example, POYO+ outperformed pseudosubjects of 30 neurons by more than 10% and 200 neurons by almost 20%. This suggests that POYO+ is able to extract latent information with much fewer neurons than CEBRA.

# E  ANALYSIS OF LATENT EMBEDDINGS

We provide details about the process of analysis to obtain Figure 4.

## E.1  EXPERIMENT DETAILS

To carry out the analysis in Figure 4, we extracted the latents from a random sample of approximately 500,000 1s windows. For a 1s context window, POYO+ produces 64 latent tokens, each with a 128-dimensional embedding. We concatenated these latents, resulting in a combined 8192-dimensional

latent representation for each sample. For part of the analyses, we will compute, session-averaged, and Cre-line averaged latent representations.

**Dimensionality Reduction by t-SNE** For the purposes of visualizing the latent space, we down-sample the number of samples to around 110,000. We first used Principal component analysis (PCA) to reduce the dimensionality from 8,192 to 50. Next, we performed t-SNE (van der Maaten & Hinton, 2008) to further reduce the dimentionality from 50 to 2 for visualization.

**Linear Decoding** For all linear decoding tasks, we used the scikit-learn implementation of logistic regression with a train-validation-test split of 60/20/20. We performed hyperparameter search for regularization parameter $C$ between 0.1 and 500.. We determine the best model based on the performance on the validation set, and report the linear layer's performance on the test set. To ensure a fair comparison with the Baseline metrics, we calculated the average latent representation for each session instead of using all samples individually.

### E.2 CRE-LINE AND BRAIN REGION DECODING

Here, we report the accuracies for cell type and brain region classification, from our model, the single task (DG) model, and the the Baseline feature-based classifier in Schneider et al. (2022) (Table A4). It is worth noting that POYO (DG) only contains roughly 400 samples in total, while the Baseline and POYO+ have over 1000 samples for linear decoding.

Table A4: Performance comparison of Baseline, POYO (DG) , and POYO+ across different region and cell type classification tasks.

|  | Num of classes | Baseline | POYO (DG) | POYO+ |
|---|---|---|---|---|
| Cre-line | 13 | 56.00% | 39.57% | 67.93% |
| Cre-line + Layer | 24 | 38.48% | 16.86% | 40.88% |
| Area | 6 | 36.47% | 39.01% | 59.81% |

**Baseline Metrics** Inspired by Schneider et al. (2023a), we used the calcium event data provided by the Allen Institute to compute the inter-spike intervals (ISIs) for each neuron in each session. Using these ISIs, we calculated the following metrics for each neuron:

- Minimum, maximum, and median of ISIs
- Mean firing rate (mean FR)
- Standard deviation (SD)
- Coefficient of variation (CV)
- Local variation (LV)
- Revised local variation (LVR)
- Local alternative to the coefficient of variation (CV2)
- Spectral density power in four bands: delta, theta, alpha, beta, and gamma (PSD)
- Two parameters of the gamma distribution (a and b) used to describe the ISI distribution ($\alpha$ and $\beta$)

After calculating these metrics, we averaged the results across all neurons in each session to obtain Baseline metrics at the session level.

It is important to note that due to the sparse nature of calcium data, LV, LVR, and CV2 values cannot be calculated. As a result, these metrics are excluded from the subsequent analyses.

### E.3 LEARNED TASK EMBEDDINGS

In Figure A4, we visualize the learned task embeddings in POYO+ . We find that related tasks typically cluster together. For example the various drifting gratings decoding tasks, or the decoding tasks corresponding to the three natural movies.

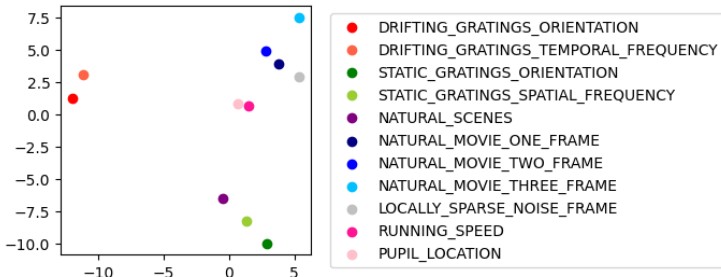

Figure A4: PCA projection of learned task embeddings

### E.4 VISUALIZATION OF MORE FINE-GRAINED CRE-LINE CONFUSION MATRIX

We provide a visualization of the confusion matrix for a more fine-grained classification task with POYO+ (Figure A5).

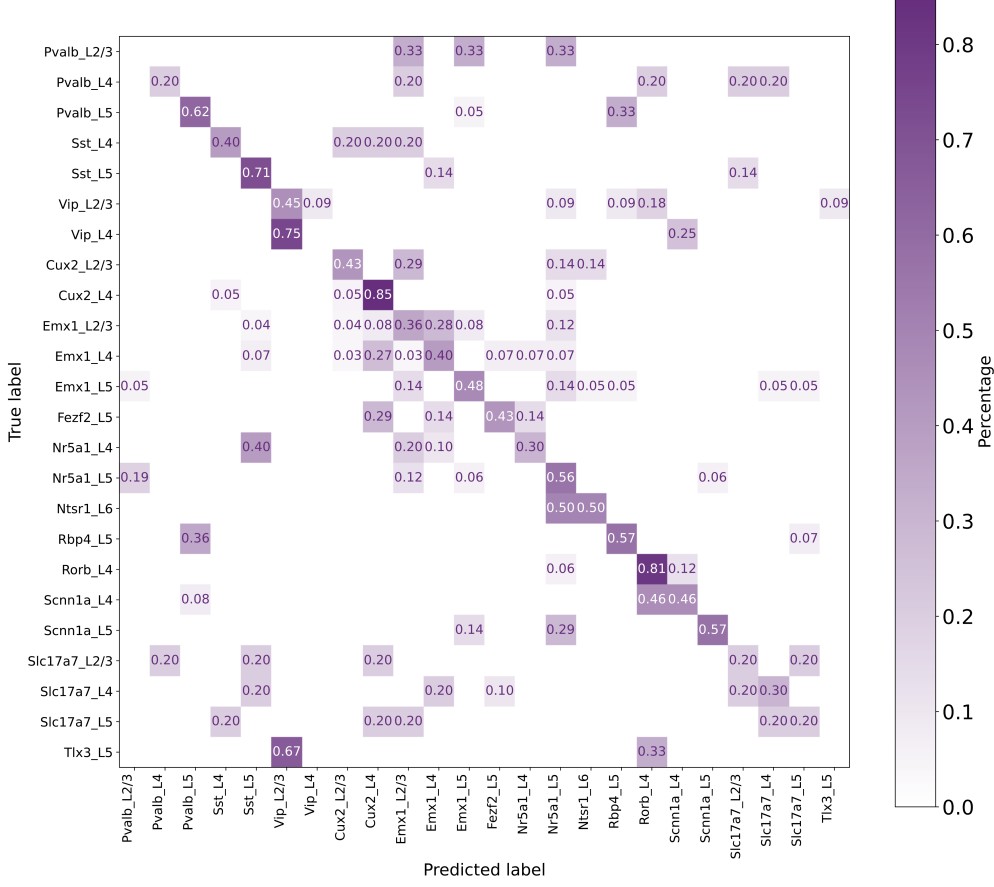

Figure A5: **Examining the organization of the latent embeddings from POYO+ by Cre-line.** Confusion matrix for Cre-line classification, including the layer for each Cre-line.

### E.5 VISUALIZATION OF BASELINE AND POYO (DG)

We provide the confusion matrices for Baseline and POYO (DG) in classification tasks (Figure A6, Figure A7).

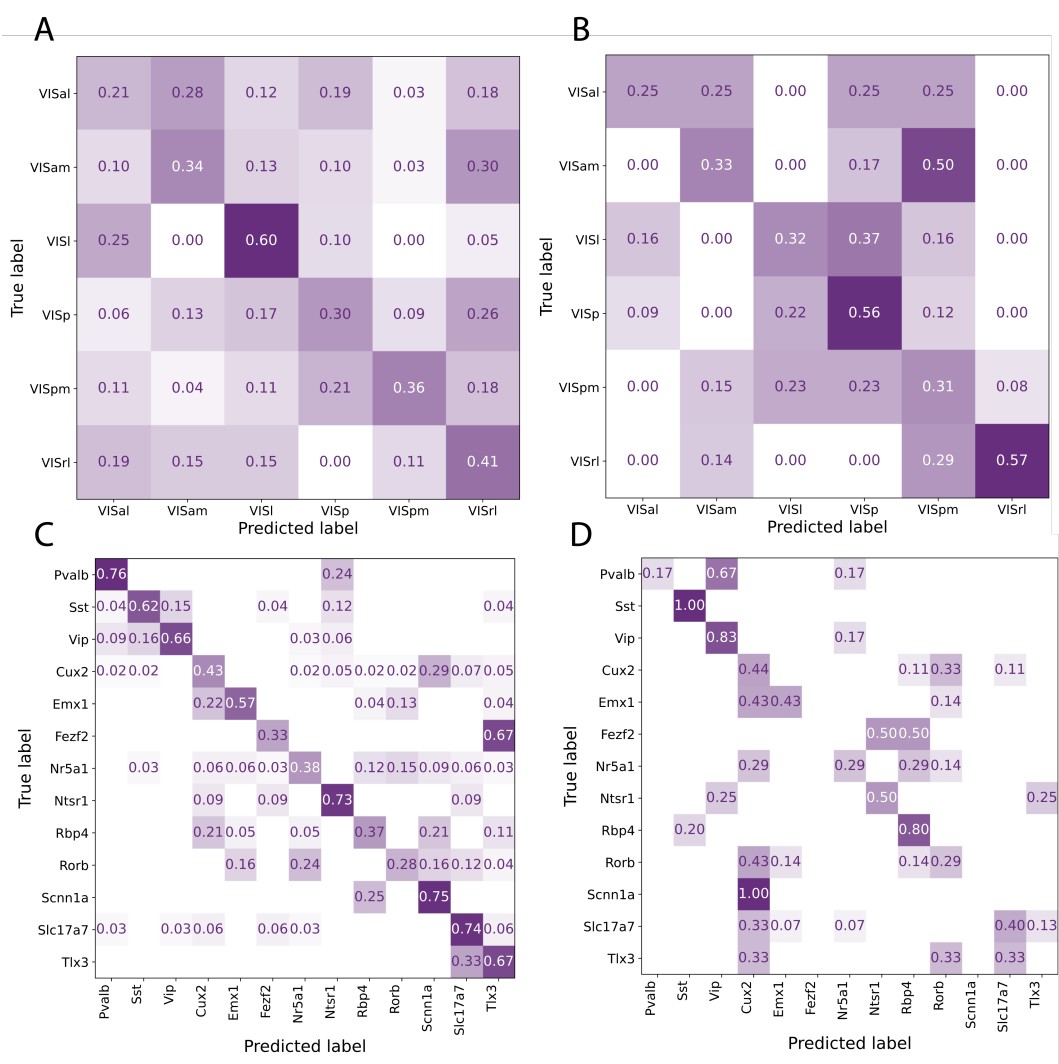

Figure A6: **Baseline and POYO (DG) confusion matrices for brain area and Cre-line classification.** A. Brain area classification (Baseline). B. Brain area classification (POYO (DG) latents). C. Cre-line classification (Baseline). D. Cre-line classification (POYO (DG) latents).

We also provide the dendrograms for detailed cre-lines and brain area for POYO (DG) (Figure A8).

E.6  TASK DEPENDENT MODULATION OF EMBEDDINGS BY CRE-LINE

We provide the extended version of Figure 5 in this subsection (Figure A9).

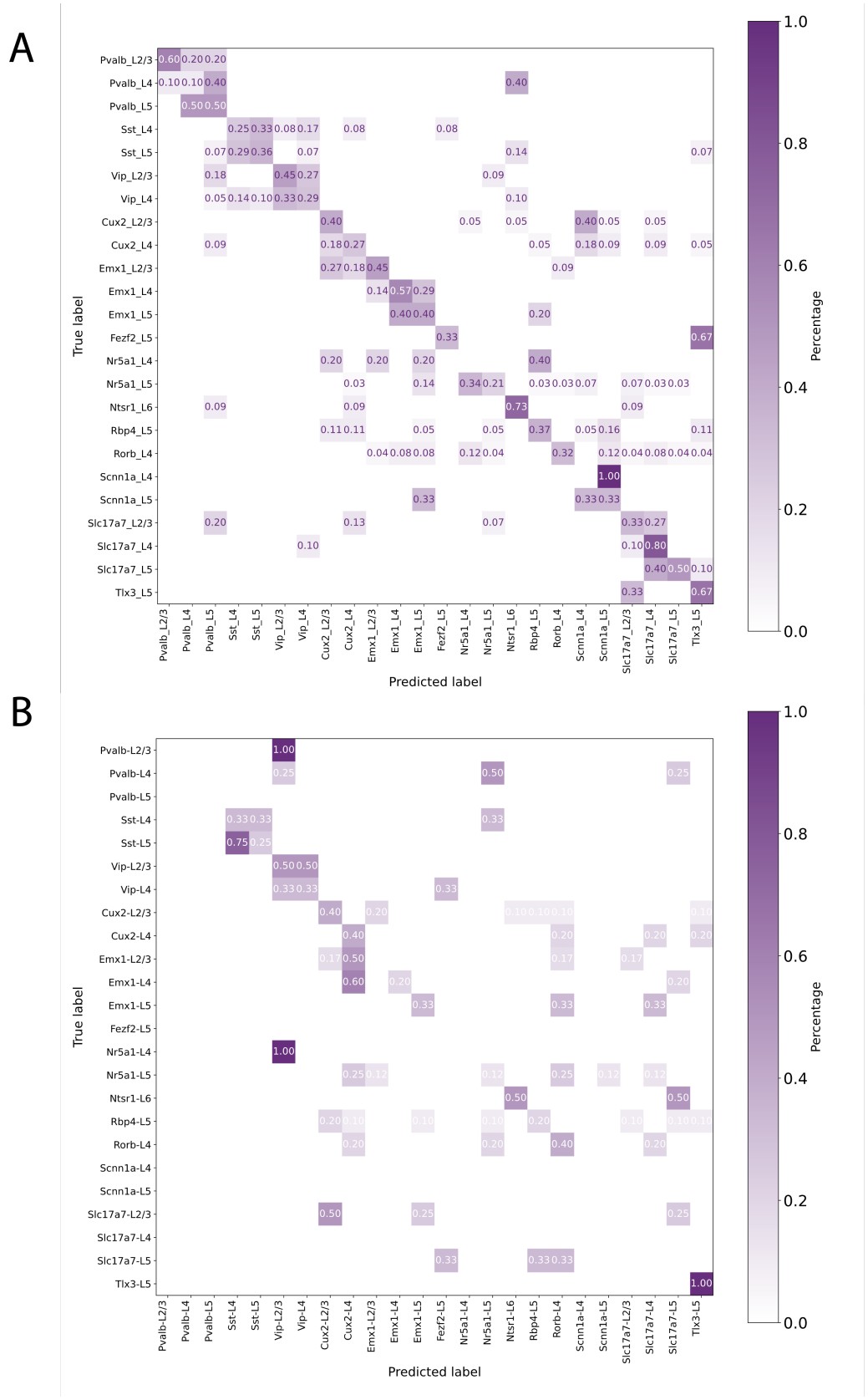

Figure A7: Baseline and POYO (DG) confusion matrices for fine-grained Cre-line + layer classification. A. Baseline. B. POYO (DG) .

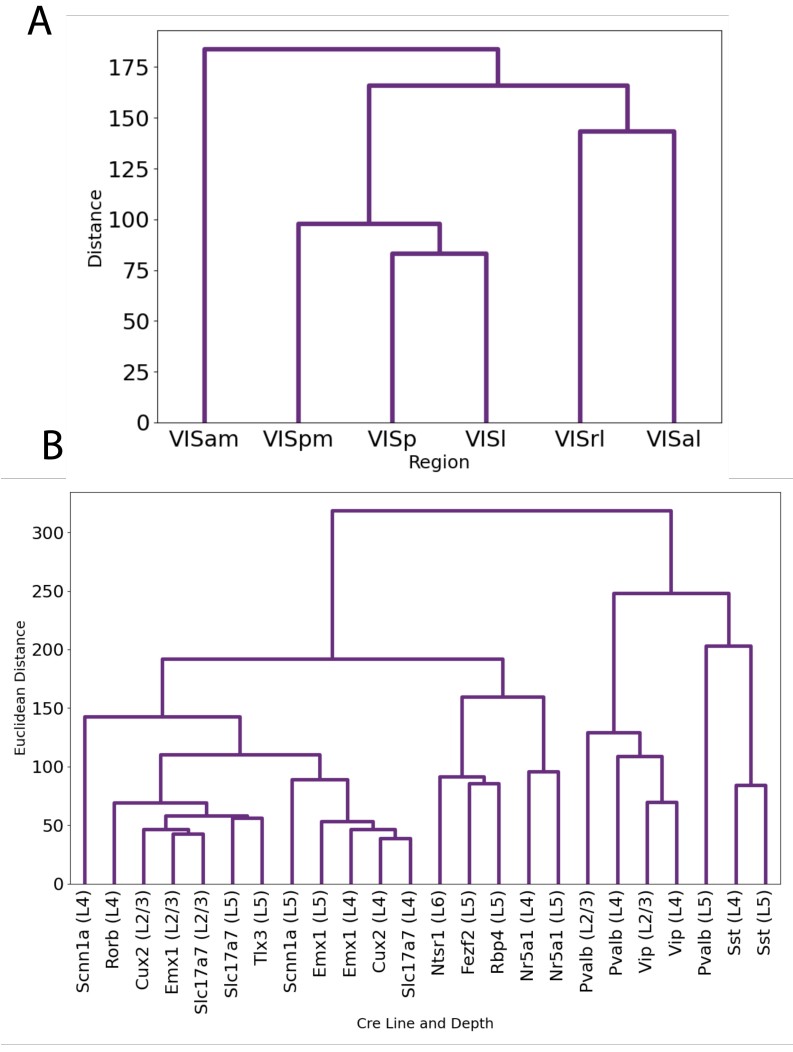

Figure A8: Visualizations of hierarchical clusterings from POYO (DG) . A. Brain areas. B. Cre-lines.

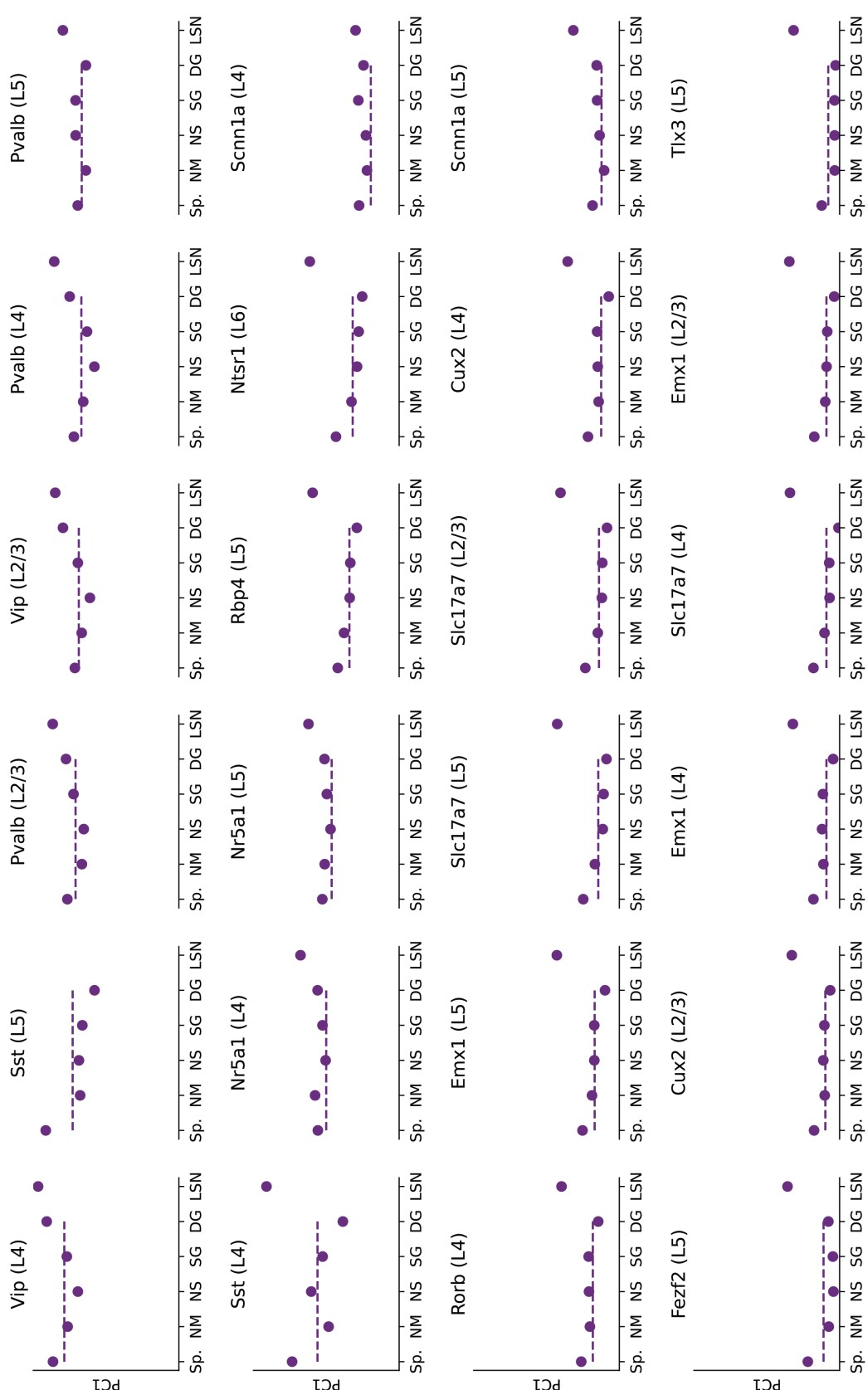

Figure A9: Deviation of cre-line averaged embeddings from Baseline.

