# OpenReview forum: "Multi-session, multi-task neural decoding from distinct cell-types and brain regions"
_ICLR.cc/2025/Conference — ICLR 2025 Spotlight_

### Official Review · Reviewer_RcB8 · 2024-11-02

**Soundness:** 2
**Presentation:** 3
**Contribution:** 2
**Rating:** 6
**Confidence:** 4

**Summary:**

This study investigates the impact of data scale on the performance of brain decoding models, developing a multi-task transformer architecture trained on the brain observation dataset from the Allen Institute for Brain Science. This dataset includes response data from over 100,000 neurons across six brain regions in mice, demonstrating the effects of different stimuli on neuronal activity. The research finds that transfer learning across brain regions and cell subtypes is effective, and that integrating data from heterogeneous datasets can enhance decoding performance. Despite the heterogeneity of neural circuits, the model's latent representations can automatically distinguish between different brain regions and cell subtypes, indicating that the shared computational principles among them can be extracted. The main contributions of the research include: 1) the introduction of a novel multi-task decoder capable of flexibly handling various decoding tasks; 2) validation of successful transfer learning across brain regions and cell types; 3) provision of evidence that increasing the diversity of brain regions can improve decoding performance; and 4) demonstration of meaningful structures in the model's latent representations that reflect biological differences. Overall, this work offers new insights into the similarities and differences among heterogeneous neural circuits and lays the groundwork for training large-scale neural decoding models.

**Strengths:**

This paper presents a novel multi-task transformer architecture for decoding neural activity from a large dataset comprising over 100,000 neurons across six different brain regions.  The originality lies not only in the innovative architecture itself but also in the approach of leveraging transfer learning across heterogeneous datasets.  By demonstrating that insights gained from one brain region can enhance decoding performance in another, the authors push the boundaries of existing methodologies in neural decoding. The quality of the research is commendable, with a robust experimental setup that includes training on extensive neural response data.  The authors employ rigorous validation techniques and comprehensive analyses, providing a solid foundation for their claims.  The results are quantitatively backed by clear metrics that illustrate the performance improvements achieved through their proposed model, showcasing its practical applicability. Generally, this findings not only contribute to our understanding of neural dynamics but also propose a framework that can be utilized in various applications, ranging from neuroprosthetics to brain-computer interfaces.

**Weaknesses:**

While the authors provide a comprehensive analysis of their proposed multi-task transformer architecture, the experimental validation could be further strengthened.   Specifically, the comparison against state-of-the-art methods in the field is somewhat limited.   For a more robust assessment of the proposed model's efficacy, additional benchmarks should be included, such as comparisons with other established neural decoding frameworks.   This would provide clearer context for the reported performance improvements and demonstrate the advantages of the proposed approach more convincingly.  Meanwhile, this study relies heavily on a large dataset from six brain regions;   however, the diversity of this dataset may not sufficiently represent the variability found across different species or experimental conditions.   To enhance the generalizability of the model, the authors could consider including additional datasets that encompass various neural activity patterns from other organisms or contexts.   This would help validate the transfer learning claims and ensure that the model performs well across a broader spectrum of neural encoding scenarios.
Furthermore, the discussion of limitations is relatively sparse.   A more thorough acknowledgment of potential shortcomings, such as the influence of noise in the data or the computational demands of the proposed model, would provide a more balanced perspective.   Including these considerations would demonstrate the authors' awareness of the challenges and offer avenues for future research to address these limitations effectively.

**Questions:**

The authors mention a data-driven feature selection process.  Could they elaborate on the specific criteria and methods used for selecting features from the neural data?

The evaluation metrics presented in the paper primarily focus on accuracy and predictive performance.  Can the authors justify their choice of metrics, particularly in relation to the biological significance of the results?  Are there any additional metrics that could provide a more comprehensive assessment of the model's effectiveness, particularly in understanding the underlying neural mechanisms?

With the model's complexity and the relatively limited size of the dataset (six brain regions), what steps have the authors taken to mitigate the risk of overfitting?

---

> ### Author Response · Authors · 2024-11-22
>
> Thank you for your comments and questions, and for finding that our method “pushes the boundaries of existing methodologies in neural decoding”. In what follows, we will provide point-by-point replies to your questions.
>
> > 1. "The comparison against state-of-the-art methods in the field is somewhat limited [...] additional benchmarks should be included, such as comparisons with other established neural decoding frameworks.”
>
> **Reply:**  We compare our method to POYO which is a state-of-the-art method for neural decoding for monkey motor decoding and in the Neural Latents Benchmark. We also include other baselines including CEBRA [1], MLP, and TCN, which is still a standard [2] when it comes to neural decoding. Please let us know if you have additional neural decoding baselines that can be applied to multi-session calcium imaging data, and which you think are missing.
>
> The Allen dataset is unique in that it has both diverse cell types and areas, as well as multiple behaviors and tasks, including both continuous behaviors like pupil and running speed, along with discrete stimulus information for classification tasks. Unfortunately, there are no multi-task benchmarks that exist in the field and thus we found this dataset to be a rich source of diverse tasks and neural recording patterns.
>
> > 2. "This study relies heavily on a large dataset from six brain regions; however, the diversity of this dataset may not sufficiently represent the variability found across different species or experimental conditions. [...] the authors could consider including additional datasets [...]"
>
> **Reply:**   First, we would like to point out that the Allen Brain Observatory dataset contains activity from very diverse cell types, across a large cohort of animals. The size of this dataset is truly unprecedented in the field, and whereas most labs will collect 2-3 cre lines consistently across tens of animals, the ABO contains 3 inhibitory and 10 excitatory lines, across 256 animals, which provides a scale that cannot be captured by a single lab at a time. Our results suggest that the large-scale pretrained model provides significant benefits over the standard training paradigm, where models are trained on one animal at a time. These data are also collected under very different contexts including synthetic and naturalistic stimuli.
>
> Nonetheless, we agree that demonstrating transfer to other contexts would be helpful. Following the reviewer’s suggestion, we have now trained POYO+ on a calcium imaging dataset from the hippocampus, and shown that a model pretrained on visual cortex data can transfer to decoding position from the hippocampus. Please see Table 1 in the general reply.
>
> > 3. “Furthermore, the discussion of limitations is relatively sparse. A more thorough acknowledgment of potential shortcomings, such as the influence of noise in the data or the computational demands of the proposed model, would provide a more balanced perspective.”
>
> **Reply:** Thanks for your suggestion.  In our current discussion, we bring up a number of limitations, including the fact that “we have only tested our model in visual cortex and the current work does not incorporate other regions or species, which could further validate the generalizability of our model. Additionally, our model does not yet incorporate neuron activity prediction, which could help to further improve the model’s ability to understand the dynamics of different cellular sub-types and regions.”
> We agree that including the computational demands of the method would also be a good addition to the discussion. We will add this to the updated manuscript.
>
> > 4. “The authors mention a data-driven feature selection process. Could they elaborate on the specific criteria and methods used for selecting features from the neural data?”
>
> **Reply:**  We are not sure what the reviewer is referring to, as we do not perform any feature selection in our work. Could the reviewer kindly clarify what part of the text they are referring to?  We are using neuron dropout during training as an augmentation to regularize the model – a strategy commonly used in other methods [3]. But we do not do any feature selection.

---

> ### Author Response · Authors · 2024-11-22
>
> > 5. “The evaluation metrics presented in the paper primarily focus on accuracy and predictive performance. Can the authors justify their choice of metrics, particularly in relation to the biological significance of the results? Are there any additional metrics that could provide a more comprehensive assessment of the model's effectiveness, particularly in understanding the underlying neural mechanisms?”
>
> **Reply:**  In addition to reporting the accuracy of models in the different decoding tasks, we also study the representation of cell type and brain region in the latents, both of which are clearly very relevant to biology. In particular, we report the brain region and cell-type classification accuracy (both metrics that are not focused on behavior or stimulus decoding) and compare against other baselines. In addition to decoding different cell types, we also provide a hierarchical clustering in Figure 4 to visualize the relationships across different cell types and regions.
>
> > 6. “With the model's complexity and the relatively limited size of the dataset (six brain regions), what steps have the authors taken to mitigate the risk of overfitting?”
>
> **Reply:** The dataset has over 1335 sessions from 256 animals, and is trained on over 100,000 neurons. Thus, the scale of the model in the end is the largest that has been reported for calcium imaging data. To provide further evidence of the generalization of this approach, we refer the reviewer to the earlier experiment on data from mouse hippocampus.
>
> ---
>
> [1] Schneider et al., Learnable latent embeddings for joint behavioural and neural analysis. Nature 2023
> [2] Glaser et al., Machine Learning for Neural Decoding. eNeuro 2020
> [3] Keshtkaran et. al. A large-scale neural network training framework for generalized estimation of single-trial population dynamics. Nature Methods, 2022.

---

> ### Author Response · Authors · 2024-12-03
>
> We believe that we have addressed the reviewer’s concerns in our rebuttal. Please let us know if you have any further questions or concerns that we can address.

---

### Official Review · Reviewer_VpA8 · 2024-11-02

**Soundness:** 3
**Presentation:** 4
**Contribution:** 3
**Rating:** 8
**Confidence:** 4

**Summary:**

The authors proposed a study that extends POYO, a method for decoding cell-types and brain regions according to neural recordings, to multi-session data and multiple tasks. The authors proposed a multi-task decoder and a training strategy that used multi-session data. They also demonstrated the merits of multi-session and multi-task training through extensive experiments and comparisons.

**Strengths:**

The idea of multi-session and multi-task decoding of cell-types and brain regions is not entirely new, however, the authors proposed several key components to realize such an idea. The study was clearly presented and the validations are extensive.

**Weaknesses:**

Some details can be further elaborated. Specifically, the architecture of the multi-task decoder can be presented with more details, e.g., the numbers of self-attention blocks and attention heads.

**Questions:**

1. It is better to provide a figure to illustrate the multi-task decoder, maybe in Appendix .
2. How is the multi-task decoder proposed in this study different from a general multi-task transformer decoder? What are the unique designs in this multi-task decoder and how are those designs related to the unique characteristics of the decoding tasks?
3. “The losses are summed and weighted...”, how to determine the weight? Is it inverse proportional to the number of predictions?
4. The proposed model outperforms CEBRA in natural the movie frame decoding task, how about other tasks?
5. There are 12 single-task variants of POYO, but it seems that Fig. 3C reports 9 cases. Why are the results for the other three cases missing?
6. “ we find that the model trained on inhibitory Cre-lines does surprisingly well on the drifting gratings task.” How about other tasks?

Minor issues:
Both POYOplus and POYO+ were used in the manuscript (e.g., Fig.3). Please try to make it consistent.
Typos: “one for each mouse, imaging depth, and visual area. from various depths...”
How such relatively low decoding accuracies can support neuroscience research?

---

> ### Author Response · Authors · 2024-11-22
>
> Thank you for your comments and questions, and for appreciating our extensive experiments and analyses. In what follows, we will provide point-by-point replies to your questions.
>
> > 1. “The architecture of the multi-task decoder can be presented with more details, e.g., the numbers of self-attention blocks and attention heads.”
>
> **Reply:** We have extended Appendix C.2, to provide more details about the hyperparameters used in our experiments. We also note that we plan to release code for training and finetuning POYO+.
>
> > 2. “It is better to provide a figure to illustrate the multi-task decoder, maybe in Appendix.”
>
> **Reply:** Thank you for the suggestion, we have added Figure A1, illustrating the internals of the multi-task decoder block in Appendix C.3.
>
> > 3. “How is the multi-task decoder proposed in this study different from a general multi-task transformer decoder? What are the unique designs in this multi-task decoder and how are those designs related to the unique characteristics of the decoding tasks?”
>
> **Reply:** In literature we find two common multi-task decoder designs. In GATO, the decoder is autoregressive, the sequence starts with a task token, then the decoder autoregressively generates the sequence of tokens corresponding to the task. Typically, we will find that these decoders assume a specific frequency and structure for the data.  We design the decoder to be able to output behaviors that are otherwise sampled at different rates, this can be the case for different datasets even for the same task. This is done by using absolute timestamps in the query as opposed to using an arbitrary ordering of tokens in the sequence.
>
> Second, our decoder is designed to be scalable. Instead of having a separate decoder branch for each task, we share the same cross-attention layer across all tasks (as can be now seen in the added Figure A1). Only the last layers (linear) are task-specific because they are needed to map to the task’s target dimension.
>
> > 4. ““The losses are summed and weighted...”, how to determine the weight? Is it inversely proportional to the number of predictions?”
>
> **Reply:** Indeed, the weight for a given task is inversely proportional to the number of predictions in a given batch. We also apply a rescaling operation to rebalance the losses across tasks. The main issue we are trying to fix is the discrepancy between the regression tasks and the classification tasks. Typically, mean squared error losses can hover around 0.1 and lower, while cross-entropy losses can be high (for example 2.0). If we do not balance the losses, we will find that the model will focus on classification tasks and ignore regression tasks. To correct this issue, we let the model train for a few steps, record the scale of the losses, and then reweight the tasks to rebalance the losses across all tasks. We refer the reviewer to Table A2 in the appendix for the weights we used for the different tasks.
>
> > 5. “The proposed model outperforms CEBRA in the movie frame decoding task, how about other tasks?”
>
> **Reply:**   In the submission, we report the accuracy of multi-session CEBRA on drifting gratings and the pseudopopulation strategy for single-session for natural movie frame decoding explored in their paper. In both cases, we significantly outperform CEBRA.
>
> > 6. “There are 12 single-task variants of POYO, but it seems that Fig. 3C reports 9 cases. Why are the results for the other three cases missing?”
>
> **Reply:**  The 10th task is the one reported in Fig. 3B. We left out two tasks that we are not showing in Fig 3C, in an attempt to not overload the figure – we intended to include these in the appendix but clearly did not. The first is natural movies 2 (which is another variant of natural movies 1 and 3), the second is locally sparse noise which is a classification task that involves 4000 unique patterns. We will update the figure to include these two tasks.
>
> > 7. ““ we find that the model trained on inhibitory Cre-lines does surprisingly well on the drifting gratings task.” How about other tasks?”
>
> **Reply:** Each entry in Tables 1 and 3 involves 30 fine tuning runs. The reason we did not test on additional tasks, is that it would require running thousands of models, which is computationally restrictive. We chose to dive deeper into differences between cre-lines across tasks in Section 3.3, where we analyze the latent embeddings of POYO+. We do however agree with the reviewer, that conducting more transfer experiments for different tasks would be insightful.

---

> ### Author Response · Authors · 2024-11-22
>
> > 8. “Minor issues: Both POYOplus and POYO+ were used in the manuscript (e.g., Fig.3). Please try to make it consistent. Typos: “one for each mouse, imaging depth, and visual area. from various depths...”
>
> **Reply:** Thank you for pointing this out, we have updated the text, and Figure 3.
>
> > 9.  How such relatively low decoding accuracies can support neuroscience research?”
>
> **Reply:** The decoding accuracies we are looking at are averaged across thousands of populations of different sizes and from different areas and cre-lines. In reality, the variance is quite high. As we can see in Figure 3A, the performance of the model can be as low as 12.5% (chance) and as high as 100%. Naturally we find that the number of neurons, the brain region and the cre-line all are factors in determining whether decoding performance will be high or low. Decoding is possible at higher accuracies, it simply depends on the nature of the recording site. Additionally, we show that supporting neuroscience research is not only done through improving decoding accuracy but also through providing tools for building insights from large amounts of data, as we demonstrate in our analyses.

---

> > ### Comment · Reviewer_VpA8 · 2024-11-25
> >
> > Thank the authors for their effort in addressing my concerns. All my concerns have been adequately addressed.

---

### Official Review · Reviewer_feML · 2024-11-02

**Soundness:** 3
**Presentation:** 3
**Contribution:** 4
**Rating:** 8
**Confidence:** 3

**Summary:**

The paper introduces a novel tokenization method and multimodal neural decoder to decode behavioral and stimulus information from calcium responses recorded from multiple regions of the mouse visual cortex. The proposed model achieved state-of-the-art decoding performance in the Allen Brain Dataset; it outperforms previous models in cross-session, cross-region, and cross-dataset settings. Moreover, the clustering of cell types and brain regions emerges from the latent representation of the model purely from response-task pairs. These results suggest that it is beneficial to train neural decoders on multi-task and multi-session recordings.

**Strengths:**

- This paper demonstrates training a neural decoder on large-scale multi-session, multi-region, and multi-task recordings can greatly improve model decoding performance across a wide range of scenarios, and is an important step towards a foundation model of neural data, a topic of great interest in NeuroAI.
- The experiments are thorough, covering a wide range of realistic transfer learning scenarios.
- The analysis of the latent embedding on the trained POYO encoder is very interesting. It shows that the model, purely trained on decoding tasks, without any information on the neurons (location, type, region, etc.), learns cell-specific properties.
- Overall, the paper is clear and well-written.

**Weaknesses:**

I don’t think the paper has any major weaknesses, however, there are several points I hope the authors can clarify regarding the model setup. Please find them in the Questions section below.

**Questions:**

- What is the difference between the query tokens and output tokens in Figure 1? My understanding is that the decoder takes latent representation from the POYO encoder and the query tokens. The query tokens inform/query the decoder which task(s) to predict (e.g. running speed, pupil dilation, etc.), and then a task-specific linear layer processes the output tokens to generate the prediction. However, in Figure 1, the color code suggests the output tokens are used as query tokens. Can the authors clarify this point (and perhaps update the caption of Figure 1)?
- Following the previous question, does the model predict in an autoregressive fashion? i.e. predict one time-step at a time. Or does the model input the entire calcium trace and predict the whole segment at once? If it is the latter, is causal attention used so that the model doesn’t have access to future time-steps?
- In Section 3.1 and Figure 3, the authors compare single-session trained baselines vs multi-session trained POYO+ (also POYO and CEBRA). Do the authors have a sense of how data-hungry POYO+ is? For instance, would POYO+ outperform the baseline models in a single-session setting?
- When comparing the models' performance (Figure 3, Table 1, 2, and 3), can the authors report s.e.m. or standard deviation across (one of) trials/sessions/animals so that the readers can have a sense of how much variation is in the performance?
- In Section C.2 Training details, can the authors clarify that “the weight for each task is determined based on the scale of the loss from an initial model”? Does this mean that an untrained (randomly initialized) model is used to make predictions for each task, and then their loss magnitude is used to determine the loss scale?
- A very minor comment: I don’t think “POYO+” was introduced in the main text as the name of the method.

---

> ### Author Response · Authors · 2024-11-22
>
> Thank you for your comments and questions, and for finding that our work represents an “important step towards a foundation model of neural data”. In what follows, we will provide point-by-point replies to your questions.
>
> > 1. “What is the difference between the query tokens and output tokens in Figure 1? [...]”
>
> **Reply:**  The reason there is a parallel between query tokens and output tokens is because there is a one-to-one correspondence between each query token and each output token.
>
> The query token is not asking the question “what is the running speed?”, it is more specific, it asks “what is the running speed at time t for this session”. Meaning that we will have as many queries as there are points in the output. For example, when predicting running speed, given a context window of 1s with a sampling rate of 30fps, we will have 30 query tokens. Each token contains the information about: 1) the task 2) the session_id 3) the timestamp at which we want to decode the variable. And hence each query token will be “answered” by the decoder and the output will be the output token. We will make this more clear in the text.
>
> > 2.  “does the model predict in an autoregressive fashion? i.e. predict one time-step at a time. Or does the model input the entire calcium trace and predict the whole segment at once? If it is the latter, is causal attention used so that the model doesn’t have access to future time-steps?”
>
> **Reply:** The model does not predict the data in an autoregressive fashion. It predicts the output all at once, this would make it more similar to BERT-style prediction rather than GPT-style prediction. We do not need to use causal masking because the model takes as input the neural activity in a context window [t1, t2] and predicts behavior in the same context window [t1, t2].
>
> > 3. “[...] Do the authors have a sense of how data-hungry POYO+ is? For instance, would POYO+ outperform the baseline models in a single-session setting?”
>
> **Reply:**  We have benchmarked POYO+ in the single-session setting, when we conducted our experiments comparing CEBRA and POYO+ using the pseudo-population method in Appendix D2. We show that POYO+ can be trained on one session worth of data and outperforms the single-session CEBRA baseline. We have not been able to benchmark single-session POYO+ on the full Allen Brain Observatory dataset due to computational limitations (thousands of small models to fit). We do however note that we have multiple POYO+ models trained on various numbers of sessions (ranging from 100 to 1300 sessions) (Table 1, Table 2) where we show that the model outperforms the MLP baseline.
>
> > 4. “[...] can the authors report s.e.m. or standard deviation across (one of) trials/sessions/animals so that the readers can have a sense of how much variation is in the performance?”
>
> **Reply:** We will update Figure 3 to include error bars. We will also include extended versions of the Tables in the Appendix with standard deviation across sessions. We agree that showing the variation in performance is really important. In fact, we show this in Figure 3A, where we can see that the performance across sessions wildly varies going from 12.5% (chance) all the way to 100%. We find that the number of neurons, the brain area and the cre-line all dictate how decodable a given population is.
>
> > 5. “In Section C.2 Training details, can the authors clarify that “the weight for each task is determined based on the scale of the loss from an initial model”? Does this mean that an untrained (randomly initialized) model is used to make predictions for each task, and then their loss magnitude is used to determine the loss scale?”
>
> **Reply:** The latter statement is correct. The main issue we are trying to fix is the discrepancy between the regression tasks and the classification tasks. Typically, mean squared error losses can hover around 0.1 and lower, while cross-entropy losses can be high (for example 2.). If we do not balance the losses, we will find that the model will focus on classification tasks and ignore regression tasks. To correct this issue, we let the (randomly initialized) model train for a few steps, record the scale of the losses, and then reweight the tasks to rebalance the losses across all tasks. In practice, we found that we do not have to be very accurate when picking the weights – in the above example, we simply need to multiply the regression loss by 10 to make it in the same order of magnitude as the other loss. We refer the reviewer to Table A2 in the appendix for the weights we used for the different tasks.
>
> This approach of selecting weights is unfortunately common in other domains, including for language models where different sources of data are weighted differently based on heuristics.
>
> > 6. “A very minor comment: I don’t think “POYO+” was introduced in the main text as the name of the method.”
>
> **Reply:** We have updated this. Thank you.

---

> > ### Comment · Reviewer_feML · 2024-11-26
> >
> > > Reply: We will update Figure 3 to include error bars. We will also include extended versions of the Tables in the Appendix with standard deviation across sessions. We agree that showing the variation in performance is really important. In fact, we show this in Figure 3A, where we can see that the performance across sessions wildly varies going from 12.5% (chance) all the way to 100%.
> >
> > If the performance varies significantly, I suggest replacing the bar plots in Figures 3b and 3c (and other performance bar plots) with box plots instead. Please also report their statistical significance against the baseline.
> >
> > I thank the authors for their detailed responses which have satisfyingly addressed all my questions.

---

### Official Review · Reviewer_1QiM · 2024-11-04

**Soundness:** 3
**Presentation:** 2
**Contribution:** 2
**Rating:** 8
**Confidence:** 3

**Summary:**

The paper explores large-scale, multi-task neural decoding across different cell types and brain regions. It introduces a multi-task transformer model, named POYO+, trained on data from the Allen Institute’s Brain Observatory dataset. The paper aims to determine whether transfer learning and multi-task decoding are feasible in the paper of neural heterogeneity, where each cell type and brain region exhibits distinct response dynamics.

**Strengths:**

* Calcium Data: From what I can tell, neural decoding with calcium data is rare. Scaling calcium data could have benefits.
* Interpretability: The paper includes several interpretability analyses to show learned features in the latent representation space. This is useful for neural decoding, which should be more than just performance driven but also have scientific value.
* Transfer: The authors demonstrate that Poyo+ could be applied to other datasets as well, although I would have liked to see a dataset with different regions instead.

**Weaknesses:**

* Distinct Regions: I found the region coverage in the paper to be rather weak, only focusing on mice visual cortex. Although multiple sub-regions are considered as well, other papers use multi-region to mean coverage of the whole brain such as [1] or [2]. If the claim of novelty comes from handling diverse brain regions, then I would expect a dataset that spans multiple, functionally distinct areas of the brain. Could the authors explain why they only used the visual cortex? Is it because calcium data is difficult to collect across wider regions? If so, could the authors discuss the application of their findings to a broader set of regions? Also I’m not sure I quite buy the huge distinctions covered in Appendix A -- neural recording datasets can incorporate much larger amounts of spatial coverage.
* Experimental Limitations
    * More broadly, only using one dataset is a bit weak to clearly establish benefits from scaling. I would appreciate it if an additional more substantial experiment was studied on the OpenScope dataset or something manageable with new regions.
    * Moreover, I found figure 3 to show that Poyo+ and Poyo were rather comparable despite needing to train 12 variants of Poyo. Could the authors discuss these findings?

Overall, the paper is well written but I’m not entirely convinced of the claims. I think more discussion can go into the findings. I would raise my scores after seeing a deeper description of using distinct regions and some more focus on added experiments/limitations.

[1] Chau, Wang, et. al. Population Transformer: Learning Population-level Representations of Intracranial Activity. arxiv, 2024.
[2]  Zhang et. al. Brant: Foundation Model for Intracranial Neural Signal, NeurIPS, 2023.

**Questions:**

* I’m really surprised that CEBRA does worse than an MLP in Figure 3B. Why would this happen?
* Neuron Dropout: The paper uses a neuron dropout to remove some neurons during training. Is this normal? I couldn’t tell from CEBRA overall.

---

> ### Author Response · Authors · 2024-11-22
>
> Thank you for your comments and questions, and for appreciating our interpretability analyses. In what follows, we will provide point-by-point replies to your questions.
>
> > 1.  “Distinct Regions: I found the region coverage in the paper to be rather weak, only focusing on mice visual cortex. Although multiple sub-regions are considered as well, other papers use multi-region to mean coverage of the whole brain such as [1] or [2]. If the claim of novelty comes from handling diverse brain regions, then I would expect a dataset that spans multiple, functionally distinct areas of the brain. Could the authors explain why they only used the visual cortex? Is it because calcium data is difficult to collect across wider regions? If so, could the authors discuss the application of their findings to a broader set of regions? Also I’m not sure I quite buy the huge distinctions covered in Appendix A -- neural recording datasets can incorporate much larger amounts of spatial coverage. More broadly, only using one dataset is a bit weak to clearly establish benefits from scaling. I would appreciate it if an additional more substantial experiment was studied on the OpenScope dataset or something manageable with new regions.”
>
> **Reply:**
> This is an important issue to clarify, and we thank the reviewer for raising it. The reviewer is correct that the Allen Observatory data contains recordings from multiple visual regions, whereas other papers (such as those cited [1] and [2]) train a model on intracranial recordings that cover many different brain regions at once. However, there is a key difference between the transfer that we demonstrate here (Table 1) and the sort of multi-region training done in [1] and [2]. Specifically, in those papers, the recordings themselves are multi-region, i.e. the data being collected comes from multiple regions during a single recording session. The models are then trained on this data in one go, i.e. they are exposed to all of the regions’ data at once. In contrast, what we sought to demonstrate here was that a model could be trained on a recording from a single region, then transfer to data from different regions. This is important because acquiring whole-brain data is very challenging, and the vast majority of recordings done in systems neuroscience labs (particularly calcium imaging) include data from a single region. Thus, our goal is not only to show that we can train on multiple regions at once (as they did in [1] and [2]) but also to show that if we were to train a model on data from one set of regions, we can transfer to new regions. This is what we demonstrate, and it is distinct from the contributions of [1] and [2], and related works. We will be sure to clarify this in the updated manuscript.
>
> But, as suggested by the reviewer, we have performed new experiments to verify that we can transfer to non-visual areas as well. We obtained a calcium imaging dataset from the hippocampus as mice explore differently shaped open mazes (see Lee et al. 2023). We used this data to decode the animals’ positions in the maze, and we compare a model pretrained on the Allen Observatory data and one trained from scratch. As we show in Table 1 in the general response, the model pretrained on the visual data performs better than the model trained from scratch. This confirms that our model is capable of transferring from data collected in one region to another, even if those regions are functionally very distinct.
>
> > 2.  “Moreover, I found figure 3 to show that Poyo+ and Poyo were rather comparable despite needing to train 12 variants of Poyo. Could the authors discuss these findings?”
>
> **Reply:** Yes, we do indeed find that, despite being trained on very diverse tasks, POYO+ is comparable with single-task POYO models trained on all 12 tasks. Here, we believe the answer lies in the diversity of the tasks and the fact that the models are the same size. Specifically, when we train POYO+ we are training across heterogeneous tasks, which may not always induce shared underlying neural dynamics. As such, it remains impressive that a multi-task model is possible – in many domains, generalist models may not always reach the performance of specialist models [3]. Additionally, we believe that comparing the models does not stop at behavior decoding performance; our latent analysis comparisons between POYO (DG) and POYO+ show  a huge boost in decoding accuracy of the different Cre-lines if we train on all tasks. As well, we note that POYO+ transfers better to the OpenScope data (Table 3), supporting the fact that with more diverse pretraining data, we find improved transfer and generalization.

---

> ### Author Response · Authors · 2024-11-22
>
> > 3. “I’m really surprised that CEBRA does worse than an MLP in Figure 3B. Why would this happen?”
>
> **Reply:** In a single-session setting, we find that CEBRA performs more comparably to POYO+ (see Appendix D2), which suggests that it works well in simpler scenarios. However, as we attempt to scale to the multi-session setting (~300 sessions across multiple regions and cre-lines), we found that CEBRA struggles to align data across many sessions. In comparison to the multi-session setting, the MLPs are trained and tuned individually for each session, allowing them to specialize and optimize for the specific characteristics of each dataset.
>
> > 4. “Neuron Dropout: The paper uses a neuron dropout to remove some neurons during training. Is this normal? I couldn’t tell from CEBRA overall.”
>
> **Reply:** Neuron dropout is used for regularization, and is only active during training. During training, we randomly drop a subset of neurons in the population – this subset is randomly sampled at each step. Neuron dropout is a fairly common augmentation used in previous models [4] to help regularized neural models, and prevent the model from overfitting specific neurons. CEBRA does not employ neuron dropout during training.
>
> ---
>
> [1] *Chau, Wang, et. al. Population Transformer: Learning Population-level Representations of Intracranial Activity. arxiv, 2024.*
>
> [2] *Zhang et. al. Brant: Foundation Model for Intracranial Neural Signal, NeurIPS, 2023.*
>
> [3] *Reed et. al. A generalist agent. TMLR, 2022.*
>
> [4] *Keshtkaran et. al. A large-scale neural network training framework for generalized estimation of single-trial population dynamics. Nature Methods, 2022.*

---

> > ### Comment · Reviewer_1QiM · 2024-11-25
> >
> > Thank you for the comments and additional baselines. I think the positioning of the paper makes a lot more sense and this was very useful. I would suggest all the writing changes be incorporated into the paper. I will raise my score.

---

### Author Response · Authors · 2024-11-22
**General Response**

We would like to thank the reviewers for their thoughtful feedback and questions. We are pleased to hear the reviewers appreciate the work and agree that:
- The work addresses an important topic: “An important step towards a foundation model of neural data, a topic of great interest in NeuroAI.” (feML);
- The results of the paper are significant:  “By demonstrating that insights gained from one brain region can enhance decoding performance in another, the authors push the boundaries of existing methodologies in neural decoding.” (RcB8); “The paper includes several interpretability analyses to show learned features in the latent representation space. This is useful for neural decoding, which should be more than just performance driven but also have scientific value.” (1QiM); “The analysis of the latent embedding on the trained POYO encoder is very interesting. [...] the model, purely trained on decoding tasks, without any information on the neurons (location, type, region, etc.), learns cell-specific properties.” (feML)
- The experiments are rigorous and cover a wide range of transfer scenarios: “The experiments are thorough, covering a wide range of realistic transfer learning scenarios.” (feML); “The quality of the research is commendable, with a robust experimental setup that includes training on extensive neural response data.” (RcB8). “The study was clearly presented and the validations are extensive.” (VpA8)

**Thanks to the reviewers feedback, we have conducted a new experiment that provides evidence that transfer to non-visual areas is possible!**

As suggested by reviewers 1QiM and RcB8, we have performed new experiments to verify that we can transfer to non-visual areas as well. We obtained a calcium imaging dataset from the hippocampus as mice explored differently shaped open mazes [1]. We used this data to decode the animals’ positions in the maze. We compared a POYO+ model pretrained on the Allen Observatory data and one trained from scratch. Importantly, for the pretrained model, we froze most of the weights when training on 10 sessions worth of hippocampus data, modifying only the cross-attention layers at the input and output. As we now show in the Table below, the model pretrained on the visual data performs better than the model trained from scratch. Moreover, both models are better in accuracy than the Bayesian decoding model applied by [1], which produced a position decoding error of ~12 cm (see Fig. 1F in [1]). This confirms that our model is capable of learning representations that transfer from data collected in one region to another, even if those regions are functionally very distinct. We will include these results in the Appendix in the updated version of the manuscript.

| Method                  | Euclidean Distance |
|-------------------------|---------------------|
| POYO+ from scratch    | 10.33 cm           |
| POYO+ pre-trained   | 9.58 cm            |
*Table 1: POYO+ transfer to Lee et al. 2023 dataset [1]*
---
[1] Lee et al., Identifying representational structure in CA1 to benchmark theoretical models of cognitive mapping. Neuron 2023

---

### Author Response · Authors · 2024-12-03

We thank the reviewers for their responses and for engaging in the discussion! We believe that our work is now stronger thanks to all of the reviewer feedback.

---

### Meta-Review · Area_Chair_dh2f · 2024-12-08

**Metareview:**

This work tackles the difficult problem of developing general purpose brain decoding models that can overcome the large levels of heterogeneity in the brain. The proposed approach the authors take combine computational advances in transformers, along with data availability advances from the Allen Institute to provide the data needed to train such models. The authors then demonstrate their method on real datasets, comparing to relevant prior art. While the reviewers had some suggestions for improvements, no major weaknesses were noted. I therefore recommend accepting this paper.

**Additional Comments On Reviewer Discussion:**

The extend of the reviewer discussion was that some reviewers felt satisfied with the added content addressing the minor issues, which resulted in increased scores.

---

### Decision · Program_Chairs · 2025-01-22

Accept (Spotlight)